# Major sources of North Atlantic Deep Water in the subpolar North Atlantic from Lagrangian analyses in an eddy-rich ocean model

Jörg Fröhle[1,2,*], Patricia V. K. Handmann[1,*], and Arne Biastoch[1,2]

[*]These authors have contributed equally to this work and share first authorship
[1]GEOMAR Helmholtz Centre for Ocean Research, Kiel, Germany
[2]Christian-Albrechts-Universität zu Kiel, Kiel, Germany

**Correspondence:** Jörg Fröhle (jfroehle@geomar.de), Patricia Handmann (phandmann@geomar.de)

**Abstract.** The North Atlantic Deep Water (NADW) is a crucial component of the Atlantic Meridional Overturning Circulation and, therefore, is an important factor of the climate system. In order to estimate the mean relative contributions, sources and pathways of the NADW at the southern exit of the Labrador Sea, a Lagrangian particle experiment is performed. The particles were seeded according to the strength of the velocity field along the $53°N$ section and traced $40$ years backward in time in the three-dimensional velocity and hydrography field. The resulting transport pathways, their sources and corresponding transit time scales were inferred. Our experiment shows that, of the $30.1\,Sv$ of NADW passing $53°N$ on average, the majority is associated with diapycnal mass flux without contact to the atmosphere, accounting for $14.3\,Sv$ ($48\%$), where $6.2\,Sv$ originate from the Labrador Sea, compared to $4.7\,Sv$ from the Irminger Sea. The second largest contribution originates from the mixed layer with $7.2\,Sv$ ($24\%$), where the Labrador Sea contribution ($5.9\,Sv$) dominates over the Irminger Sea contribution ($1.0\,Sv$). Another $5.7\,Sv$ ($19\%$) of NADW cross the Greenland–Scotland Ridge within the NADW density class, where about $2/3$ pass Denmark Strait, while $1/3$ cross the Iceland–Scotland Ridge. The NADW exported at $53°N$ is hence dominated by entrainment through diapycnal mass flux and the mixed layer origin in the Labrador Sea.

## 1 Introduction

The Meridional Overturning Circulation (MOC) is the global redistribution system of heat, mass, fresh water and tracers. Water mass transformation from the upper to the lower MOC component associated with deep convective mixing (Lab Sea Group, 1998; Marshall and Schott, 1999) and diapycnal mixing (Straneo, 2006; Katsman et al., 2018; Johnson et al., 2019) is occurring in only few key regions globally, one of them being the highly complex region of the subpolar North Atlantic (SPNA). The associated density increase eventually results in a net downwelling of upper Atlantic MOC (AMOC) water in density space (Johnson et al., 2019) and thereby the formation of deep and intermediate water (Rhein et al., 2011). These water masses are then transported southward through the Deep Western Boundary Current (DWBC) (Dickson and Brown, 1994; Molinari et al., 1998) as well as the interior as part of the deep AMOC branch (Bower et al., 2009). Water mass properties of the North Atlantic Deep Water (NADW) are largely imprinted within the SPNA and the Nordic Seas, and mostly maintained farther south (Haine et al., 2008).

In observations, at $53°N$, the southern exit of the Labrador Sea, all three components of the NADW are present in the DWBC. The shallowest component, named Labrador Sea Water (LSW), is thought to be majorly formed through deep convective mixing in the Labrador Sea (Yashayaev and Loder, 2016). This water mass is regularly ventilated in winter and is defined as a low potential vorticity water mass with conservative temperatures below $4°C$ and densities between $27.70 - 28.10\ kg\ m^{-3}$ in neutral density ($\gamma_n$), $27.68 - 27.80\ kg\ m^{-3}$ in potential density ($\sigma_0$) (e.g. Pickart et al., 1997; Stramma et al., 2004; Mertens et al., 2014; Liu and Tanhua, 2021) and $36.50 - 36.94\ kg\ m^{-3}$ in potential density relative to $2,000\ m$ ($\sigma_2$) (e.g. van Sebille et al., 2011; Zantopp et al., 2017). The first lower NADW (lNADW) component is the Northeast Atlantic Deep Water (NEADW) which is modified Iceland–Scotland Overflow Water (ISOW) originating at the overflows of the Iceland–Scotland Ridge (ISR) which was modified along its spreading pathway. This water mass is featuring conservative temperatures between $2.2 - 3.3°C$ and high absolute salinities of $> 34.95\ g\ kg^{-1}$ with $\gamma_n$ between $28.00 - 28.15\ kg\ m^{-3}$, $27.80 - 27.88\ kg\ m^{-3}$ in $\sigma_0$ and $36.94 - 36.98\ kg\ m^{-3}$ in $\sigma_2$ (Hansen and Østerhus, 2000; Østerhus et al., 2001; Jochumsen et al., 2015). NEADW appears as a salinity maximum at depth in the hydrography of $53°N$ below the LSW component. The deepest lNADW component is the Denmark Strait Overflow Water (DSOW) originating at the overflow sills of Denmark Strait (DS) between Greenland and Iceland with densities $\gamma_n > 28.15\ kg\ m^{-3}$, $\sigma_0 > 27.88\ kg\ m^{-3}$ and $\sigma_2 > 36.98\ kg\ m^{-3}$ (Pickart et al., 1997; Schott et al., 2006; Liu and Tanhua, 2021). NEADW and DSOW are modified through entrainment of upper ocean water after passing the GSR and descending into the SPNA (Fogelqvist et al., 2003; Chafik et al., 2020).

From the observed mid–depth flow field (Palter et al., 2016; Fischer et al., 2018), the spreading path of the mid–depth water masses is known as follows: The ISOW flows along the eastern flank of the Reykjanes Ridge after crossing the ISR and entering the Iceland basin from the Nordic Seas. Two paths, through the Charlie–Gibbs Fracture Zone (CGFZ) and the Bight Fracture Zone (Lankhorst and Zenk, 2006; Zou and Lozier, 2016; Xu et al., 2018), connect the Iceland and Irminger basins passing the Mid–Atlantic Ridge. More recent model–based studies reveal an additional westward branch from the Iceland basin through the Reykjanes Ridge and crossing through the interior Irminger Sea towards the Labrador Sea (Xu et al., 2010; Zou et al., 2020a). The mean mid–depth circulation shows a confined current band west of the Reykjanes Ridge towards DS. South of DS it encounters the DSOW (Pickart, 1992; Dickson and Brown, 1994) and is transported farther south around Greenland and the Labrador basin through the DWBC. A confined boundary current is established along the East Greenland shelf break. Once the western boundary current (WBC) passes Cape Farewell, while being partly dispersed, the WBC refocuses along the west Greenland shelf break. At Cape Desolation, where eddies are shed towards the interior Labrador Sea (Hátún et al., 2007; Prater, 2002; Lilly et al., 2003; Rieck et al., 2018) the WBC is partly dispersed and just north of it a bifurcation of the boundary current, following the $1,500$ and $3,000\ m$ isobaths of the northwestern Labrador Sea, takes place (Cuny et al., 2002; Higginson et al., 2011; Palter et al., 2016; Fischer et al., 2018). At the coast of Labrador, the flow becomes confined in the boundary current again.

The importance of Labrador Sea convection for the strength and the variability of the AMOC still reamains unclear and is currently under debate (Lozier, 2012; Rhein et al., 2013; Yeager et al., 2021). While some studies assume a direct linkage (Marshall and Schott, 1999; Yashayaev et al., 2008; Haine et al., 2008), others corroborate the assumption that the AMOC is only minimally impacted by Labrador Sea convection (Pickart and Spall, 2007; Zou and Lozier, 2016; Lozier et al., 2019;

Petit et al., 2020). Deep convective mixing has been widely observed since the 1950s (Dietrich, 1957; Lazier, 1973) in the Labrador Sea and there are increasing observations of deep convection south of Cape Farewell and in the Irminger Sea (see Rühs et al. (2021) for extensive literature collection). The interest in understanding exactly where the transformation from upper AMOC water to lower AMOC water takes place in the SPNA has increased in recent years. Several studies, within medium- to high–resolution ocean models, have shown that additionally to the deep convective mixing, diapycnal mixing between the basin interior and the boundary currents as well as densification along the spreading pathways at the boundary currents play a crucial role for the total formation of dense deep water (e.g. Straneo, 2006; Katsman et al., 2018; Desbruyères et al., 2019; Sayol et al., 2019; Georgiou et al., 2020; Petit et al., 2020; Georgiou et al., 2021). The question of the relative importance of these sources and their respective pathways for the total deep water export towards the south and its variability is not completely clear yet.

Newer research has shown that a major volume of water is transformed along the North Atlantic Current path (Desbruyères et al., 2019). This water originates from different transformation processes, which are related to different export time scales (Le Bras et al., 2020). Hence, the very localized deep convection might only be adding a comparatively small amount of transformed water to the overall NADW volume. Additionally, the observed deep convection in the Irminger Sea increased over the past years (Våge et al., 2009; de Jong et al., 2012; Jong and Steur, 2016; Piron et al., 2016; Fröb et al., 2016; de Jong et al., 2018; Rühs et al., 2021). In contrast to the well documented southward spreading of deep water south of $45°N$ from the subpolar gyre, the dynamics of formation and subsequent spreading of NADW within the SPNA are not so well documented nor understood. In this model–based study, we present i) the relative contributions of the respective deep water sources to the NADW transport at $53°N$ and ii) the pathways and advection time scales of the connections between $53°N$ and the respective deep water sources. The methods and model used to perform the desired analyses are presented in the following section 2. Subsequently, we present the sources and pathways of each deep water particle category in section 3.1. In section 3.2 the water mass properties of the different water masses are presented. To conclude and classify the results within the current literature, the results are then controversially discussed in section 4 and our conclusions close this paper in section 5.

## 2   Data and Methods

### 2.1   Lagrangian Experiment in VIKING20X-JRA-OMIP

The model output used to conduct our Lagrangian experiment is the eddy–rich nested ocean/sea–ice model configuration VIKING20X-JRA-OMIP, as the name reveals forced by the JRA55–do forcing from 1958 to 2019 (version 1.4, Tsujino et al., 2018). See Biastoch et al. (2021) for full model description of VIKING20X and the experiment used here. It is based on the global $1/4°$ resolution grid of the Nucleus for European Modelling of the Ocean code (NEMO, version 3.6, Madec et al., 2017) and the Louvain la Neuve Ice Model (LIM2, Fichefet and Maqueda, 1997). The tripolar $1/4°$ global horizontal grid is refined in the Atlantic Ocean to $1/20°$, yielding an effective grid spacing of $\leq 5\,km$ in the SPNA. It contains 46 geopotential $z$–levels, increasing in thickness from $\sim 6\,m$ at the surface to $\sim 250\,m$ in the deepest layers. Here, daily snapshots of the three–dimensional Eulerian flow and hydrographic fields are used for the offline Lagrangian particle tracking experiment.

Biastoch et al. (2021) show that the model is reproducing the major, and regional, dynamic properties in the SPNA region, such as the strength and width of the boundary currents, the position, depth and expansion of the mixed layer (see also Rühs et al., 2021), as well as an AMOC strength comparable to observations. To conduct the offline Lagrangian particle experiment, the Python module Parcels (version 2.2.2, Delandmeter and Sebille, 2019) is utilized. Trajectories are estimated by advecting virtual particles along streamlines that are calculated from the Eulerian flow field. We hence analyze the output of the ocean model in detail through the Lagrangian particle experiment.

The domain, in which the Lagrangian particle experiment is conducted, is bounded to the north by the northern boundary of the high–resolution nest of VIKING20X (the northernmost point is $69.3°N$) and by $25°N$ to the south. The easternmost point is $20°E$, while the westernmost point is $77.5°W$. However, due to the tripolar grid of the model, the exact northern, eastern and western boundaries of the domain vary (see black dashed line in Figure 1).

### 2.1.1 Seeding strategy

Virtual particles are released at a section along the observational mooring array $53°N$ (Zantopp et al., 2017) off the coast of Newfoundland (yellow line in Figure 1), which is part of the Overturning in the subpolar North Atlantic Program (OSNAP, Lozier et al., 2017, 2019). The section in the model is approximated following the tripolar model grid in $x$- and $y$-direction (Handmann, 2019, chapter 4.3). Virtual particles are released daily during the period 2010 through 2019 in each grid box along this section. Following Schmidt et al. (2021), the amount of particles released in each grid box is defined relative to the volume transport associated with each individual grid box. If $V_{gb}$ is the volume transport of a given grid box and $V_{th} > 0$ a volume transport threshold determining the maximum absolute volume transport assigned to an individual particle, then the number of particles $N_{gb}$ released within the given grid box is defined by:

$$N_{gb} = \left\lceil \left( \frac{|V_{gb}|}{V_{th}} \right) \right\rceil \tag{1}$$

where , $\lceil \rceil$ is the ceiling function. The volume transport $V_{P_i}$ assigned to each particle $P_i$ within a grid box is then defined as:

$$V_{P_i} = \frac{|V_{gb}|}{N_{gb}} \tag{2}$$

where $i = 1...N_{gb}$. Subsequently, particles are only released in grid boxes where $|V_{gb}| > 0$. For grid boxes where $0 < |V_{gb}| \leq V_{th}$ only one particle is released, which is assigned exactly the absolute transport that is associated with the corresponding grid box. If $|V_{gb}| > V_{th}$, the transport associated with the given grid box is distributed equally among multiple particles $P_i$. Thus, each particle is associated with a pre–defined volume transport value $V_{P_i} \leq V_{th}$ that varies among particles from different grid boxes.

The release positions of the individual particles are determined by randomly distributing the particles across their corresponding grid box. Since this study is concerned with the NADW export from the Labrador Sea, particles are only released in south–eastward directed flow. Particles are seeded throughout the entire water column with a maximum volume transport per particle of $0.1\ Sv$. This results in a total of approximately $8.9 \times 10^6$ particles being released.

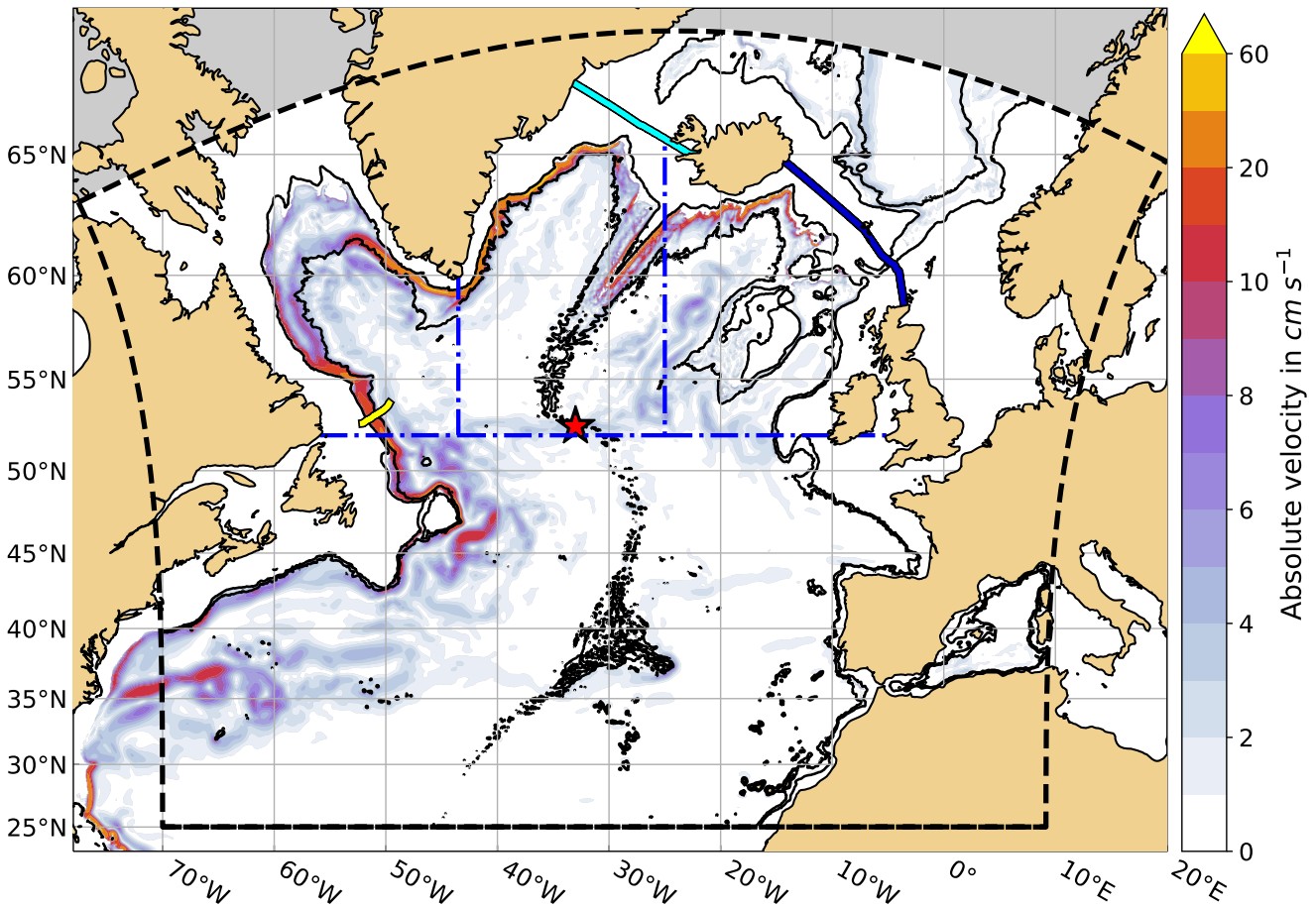

**Figure 1.** Mean absolute velocity (1958-2019, in $cm\ s^{-1}$, shading) at $1,298\ m$ in VIKING20X-JRA-OMIP. The Charlie–Gibbs Fracture Zone is indicated by the red star. The yellow line marks the $53°N$ section, the light and dark blue lines mark the Denmark Strait and Iceland–Scotland Ridge sections, respectively. The black dashed line indicates the boundary of the experiment domain considered in this study. The blue dash–dotted lines indicate the areas which are used to calculate the Labrador Sea (west of $43.5°W$; north of $52°N$), the Irminger Sea ($43.5°$ to $25°W$; north of $52°N$, south of the GSR), the Iceland basin (east of $25°W$; north of $52°N$, south of the GSR) and southern SPNA (south of $52°N$) volume transport contributions. The black contours indicate the $1,000$ and $3,000\ m$ isobaths, respectively, in the Labrador Sea area as indicated by the blue dash–dotted lines. In the remaining SPNA the black contours indicate the $1,000$ and $2,000\ m$ isobaths, respectively.

### 2.1.2 Experiment execution

The virtual particles are integrated backwards in time for $14,600$ days, or $\sim 40$ years, using a 4th order Runge–Kutta scheme at a time step of $5$ minutes. Since no additional diffusion kernel is applied, the obtained particle trajectories are equivalent to volume transport pathways (Schmidt et al., 2021). An additional kernel is however incorporated to sample potential temperature,

salinity and mixed layer depth along the particles' trajectories. Note that Parcels assumes tracer values to be constant within individual grid boxes for Arakawa C-type grids (Delandmeter and Sebille, 2019). The particle positions and properties are stored at daily resolution.

### 2.1.3 Categorisation of Particles

Here we focus on NADW, hence, only particles released within this water mass are considered during the analyses. All particles lighter than the upper NADW boundary at the seeding location are filtered out and not considered hereafter. The upper boundary of NADW is the density of the AMOC maximum at OSNAP, in VIKING20X-JRA-OMIP defined as $\sigma_{DW} = \sigma_0 = 27.62\ kg\,m^{-3}$ (Biastoch et al., 2021). To start with, the LSW was defined as $27.62\ kg\,m^{-3} \leq \sigma_0 < 27.86\ kg\,m^{-3}$. Consequently, the lNADW in the model is found at $\sigma_0 \geq 27.86\ kg\,m^{-3}$ (Handmann et al., 2018; Biastoch et al., 2021). The water mass boundaries are defined as the mean density value over the complete model output, covering 1958 through 2019. Contrary to the dynamically defined upper bound of NADW, the definition of the boundary between LSW and lNADW is based on the hydrography in the central Labrador Sea (Handmann et al., 2018). Even though, this method works fine with observations and yields the distinguished densities of the three NADW water masses, we show in this study, this does not necessarily hold for a water mass distinction in the classical sense in an ocean model. This is partly related to the unrealistically large diapycnal mixing in regions where dense waters descend topographic slopes, producing lighter water (Willebrand et al., 2001). This spurious mixing is dependent on the vertical and horizontal resolution of the ocean model and is a typical model artifact.

The particles are subsampled based on their density at their respective release, i.e only particles released at densities $\sigma_0 \geq \sigma_{DW}$ are considered, resulting in a subset of particles. These particles are referred to as **NADW$_P$**, amounting to approximately $3.5 \times 10^6$ particles. Once the particles belonging to the NADW water mass are identified, they are then divided into five mutually exclusive categories. The categories are defined based on the particles' point of origin. For each particle, the trajectory is considered only between the particle's origin, described in detail in the following, and $53°N$. Resulting from the definition of the point of origin, the trajectories have varying lengths. In turn these are consequently related to varying transit times. However, all resulting trajectories lie entirely within the NADW density range and within the North Atlantic. The terms source, origin and point of origin are used synonymous in this work. Note that despite being calculated backwards in time, the trajectories are referred to in their forward sense in the following, i.e. in flow direction. Consequently, the particle release at $53°N$ constitutes the last time step or end of the trajectory. Hence, the point of origin is considered as the first time step.

In short, each particle trajectory has a defined **point of origin**, or source. This **point of origin** is defined as the point where a particle changes its density from $\sigma_0 < \sigma_{DW}$ to $\sigma_0 \geq \sigma_{DW}$ or where it last crosses a defined section with a density $\sigma_0 \geq \sigma_{DW}$.

Particles that cross the Greenland–Scotland Ridge (GSR) and retain densities of $\sigma_0 \geq \sigma_{DW}$ represent NADW crossing the GSR from the Nordic Seas into the SPNA. The section which particles need to cross in order to be taken into account here is a combination of two subsections, the DS and the ISR. The particles are classified as **DS$_P$** or **ISR$_P$** depending on the section they cross. The subsections are extracted from the model grid as described in Handmann (2019, chapter 4.3). In Figure 1 the two sections are indicated by the light and dark blue solid lines, respectively. For DS$_P$ and ISR$_P$ the point of origin is the last crossing of the GSR within the NADW density, before reaching $53°N$. The particle information along the respective trajectories is only

considered between the last crossing of the GSR and arriving at $53°N$. Therefore, parts of the trajectories lying within the Nordic Seas or recirculating over the GSR are not considered, as we do not consider the density change north of this section. Even though, these two particle categories do not necessarily resemble the overflow water masses from observations, since all NADW is considered, we call them overflow water in the following as it is NADW flowing over the sills.

If particles increase their density during the experiment, from $\sigma_0 < \sigma_{DW}$ to $\sigma_0 \geq \sigma_{DW}$, outside of the mixed layer before reaching $53°N$, without contact to the atmosphere, this is referred to as diapycnal mass flux and the particles are classified as **DIA$_P$**. Else, if the respective density increase occurs within the mixed layer, with contact to the atmosphere, the particles are classified as **ML$_P$**. The pivotal density change of a particle is the last increase from $\sigma_0 < \sigma_{DW}$ to $\sigma_0 \geq \sigma_{DW}$ before reaching $53°N$. Hence, the exact processes and property changes in the mixed layer are not explicitly considered here. To separate

**DIA$_P$** from **ML$_P$** the particle depth is compared to the instantaneous mixed layer depth at the particle position, which is stored during the experiment along each particle's trajectory (section 2.1.2). For **DIA$_P$** trajectories are only considered between the point of respective density increase and reaching $53°N$, i.e. the point of origin is defined as the point of density transition from the upper AMOC component to the NADW below the mixed layer. For **ML$_P$** trajectories are considered between leaving the mixed layer and arriving at $53°N$, i.e. the point of origin is the location where the particles leave the mixed layer, after having

changed their density from the upper AMOC to NADW density within the mixed layer.

The particles that can not be assigned to any of the previous categories form the last category. Particles belonging to this category retain densities $\sigma_0 \geq \sigma_{DW}$ throughout their entire advection time and are referred to as **RES$_P$**. These particles either reside within the North Atlantic during the whole experiment, or enter the domain at any point in time through its lateral boundaries, except through the GSR. Particles of this category do not have a defined point of origin.

It is important to point out that every particle belonging to any of the described categories can still be entrained into the mixed layer. These mixed layer contacts are then, however, not associated with a densification from $\sigma_0 < \sigma_{DW}$ to $\sigma_0 \geq \sigma_{DW}$.

In order to differentiate the region of densification the particle categories are further divided by their position above topography. Since, in the SPNA the boundary current sticks to the strong shelf break, the particle is classified as being in the boundary if the underlying bathymetry is shallower than $3,000\,m$ in the Labrador Sea, or $2,000\,m$ in the remaining SPNA (Figure 1). If

the bathymetry is deeper, the density transition is located in the basin interior.

## 2.2   Analyses

In the following, all volume transport estimates are given with respect to the 10-year mean NADW volume transport at $53°N$ of $30.1\,Sv$ from 2010 to 2019. First, particles are grouped based on a certain condition (e.g. point of origin). Then the cumulative transport of all particles within a group is divided by the cumulative transport of all NADW$_P$. The obtained fraction is then

multiplied by the mean transport at $53°N$ to obtain the mean volume transport associated with the defined particle group.

To derive the relative and absolute ***transport contributions*** of the different volume transport sources, the particles are separated into the five categories, as described in section 2.1.3. The corresponding contributions are then estimated as explained above.

In order to compute the ***transport distribution*** at $53°N$ particles are grouped into $5\,km \times 0.01\,kg\,m^{-3}$ bins, where the distance refers to the horizontal distance from the starting point of the section. To obtain a depth profile, the transport is then summed over all distance bins and two density bins each, resulting in $0.02\,kg\,m^{-3}$ bins.

To evaluate the ***horizontal pathways*** of the particles, we follow section 4.3 in van Sebille et al. (2017). A regular $0.25° \times 0.25°$ latitude–longitude grid is defined. For each grid cell the transport–weighted number of particles visiting the grid cell is estimated, which is independent of the respective flow direction through the grid cell. Each particle, however, is only accounted for once per grid cell (recirculation is not considered). By dividing the resulting cumulative transport per grid cell by the total transport of all NADW$_P$, transport–weighted probability maps are derived. These reflect the pathways in the horizontal plane that are associated with most of the volume transport conducted by the particles. In other words, the transport–weighted fraction of particles visiting a certain grid cell at least once is obtained. Within each grid cell values can range between 0 and 1, or $0\%$ and $100\%$, equivalently. This would be the case if none ($0\%$) or all ($100\%$) particles pass through the same grid cell. In the following, grid cells with values $< 0.01\%$ are masked out.

Since, by definition, the source of DS$_P$ and ISR$_P$ is known, the point of origin is only binned for ML$_P$ and DIA$_P$. A regular $0.5° \times 0.5°$ latitude–longitude grid is defined and for each grid cell, the transport–weighted number of particles, whose point of origin is located within the particular grid cell, is obtained. Therefore, integrating the transport over all grid cells yields the volume transport at $53°N$ associated with DIA$_P$ and ML$_P$, respectively. The resulting maps can then be used to identify regions from which most of the NADW volume transport originates. Grid cells with values $< 10^{-4}\,Sv$ are masked out. To determine the transport contributions of different ocean basins, the SPNA is divided into four distinct areas. These areas are indicated by the blue dash–dotted lines in Figure 1. Integrating the transport over all grid cells within each of these defined areas yields the transport associated with the respective basin. The different regions are denoted Labrador Sea, Irminger Sea, Iceland basin and southern SPNA in the following.

For each particle the ***transit time***, i.e. the time it takes a particle, starting from its respective point of origin, to reach $53°N$ is calculated. The particles are then binned based on their ***transit times*** into 1 month bins. The transport per bin is estimated as described above. Additionally, the transport is summed over all bins, cumulatively.

To obtain the ***volumetric water mass transformations*** the particles are grouped by their water mass properties at $53°N$ and at their point of origin. The considered properties are $\sigma_0$, absolute salinity ($S_A$) and conservative temperature ($\Theta$), with bin sizes of $0.025\,kg\,m^{-3}$, $0.01\,g\,kg^{-1}$ and $0.2°C$, respectively. These properties were computed from the practical salinity, potential temperature and depth tracked along each trajectory using the TEOS–10 toolbox for Python (Intergovernmental Oceanographic Commission, 2015). The difference between the volume at $53°N$ and the volume at the point of origin for $\sigma_0$, $S_A$ and $\Theta$ class then gives the ***volumetric water mass transformation***.

To evaluate the ***evolution of the particle properties*** along the individual pathways, the maximum or minimum salinity or temperature is determined along a particle trajectory. Then the location along the trajectory is determined where the difference in salinity or temperature between the extremum and the particle source is halved. Based on these locations the particles are then grouped into $0.5° \times 0.5°$ longitude–latitude bins and the transport–weighted mean particle depth per bin is estimated.

**Table 1.** Mean transport from 2010-2019 in $Sv$ at $53°N$ associated with each particle category (DIA$_P$, ML$_P$, DS$_P$, ISR$_P$ and RES$_P$), as detailed in section 2.1.3, as well as their relative contributions in % for $\sigma_0 \geq 27.62\ kg\ m^{-3}$ (NADW), $27.62 \leq \sigma_0 < 27.86\ kg\ m^{-3}$ (LSW) and $\sigma_0 \geq 27.86\ kg\ m^{-3}$ (lNADW). The transports are rounded to $0.1\ Sv$.

| | $\sigma_0 \geq 27.62\ kg\ m^{-3}$ (NADW) | | $27.62 \leq \sigma_0 < 27.86\ kg\ m^{-3}$ (LSW) | | $\sigma_0 \geq 27.86\ kg\ m^{-3}$ (lNADW) | |
|---|---|---|---|---|---|---|
| total transport | $30.1\ Sv$ | 100% | $26.7\ Sv$ | 100% | $3.4\ Sv$ | 100% |
| DIA$_P$ | $14.3\ Sv$ | 48% | $12.8\ Sv$ | 48% | $1.5\ Sv$ | 44% |
| ML$_P$ | $7.2\ Sv$ | 24% | $7.0\ Sv$ | 26% | $0.2\ Sv$ | 5% |
| DS$_P$ | $3.8\ Sv$ | 13% | $3.4\ Sv$ | 13% | $0.4\ Sv$ | 11% |
| ISR$_P$ | $1.9\ Sv$ | 6% | $1.7\ Sv$ | 6% | $0.2\ Sv$ | 5% |
| RES$_P$ | $2.9\ Sv$ | 10% | $1.7\ Sv$ | 7% | $1.2\ Sv$ | 34% |

## 3   Results

### 3.1   Sources, Pathways and Transit Time Scales

This chapter starts with a general assessment of the absolute and relative transport contributions of the defined particle categories (section 2.1.3, Figure 1) and a description of the transport distributions (Figure 2). Common features of the particle pathways (Figure 3) are introduced as well. This is followed by a detailed evaluation of the origin locations (Figure 4) and their relative importance (Table 2), the spreading pathways (Figure 3) and the associated transit time scales (Figure 5) for each particle category, ordered by the relative contribution to the transport at $53°N$.

The mean southward NADW transport in the presented Lagrangian experiment (Figure 2 a, i) shows two peaks in density space, the first of which is located around $\sigma_0 = 27.80\ kg\ m^{-3}$. A secondary peak is found around $\sigma_0 = 27.87\ kg\ m^{-3}$ (Figure 2 i). The upper, lighter transport peak is associated with transport peaks around $27.80\ kg\ m^{-3}$ for all four defined particle sources (Figure 2 iii-vi). The dense maximum (Figure 2 i) on the other hand is dominated by DIA$_P$ and RES$_P$ (Figure 2 ii-iii). Diapycnal mass flux dominates the transport distribution throughout the water column with an overall mean transport of $14.3\ Sv$ (48%, Table 1 and Figure 2). In the density range $27.62 \leq \sigma_0 < 27.86\ kg\ m^{-3}$ DIA$_P$ amount to $12.8\ Sv$ (48%), with the second largest contributor being particles from the mixed layer ($7.0\ Sv$ or 26%, Table 1). The component with densities $\sigma_0 \geq 27.86\ kg\ m^{-3}$ (below the red dashed line in Figure 2) is dominated by DIA$_P$ with $1.5\ Sv$ (44%), while the second largest contribution, $1.2\ Sv$ (34%), is associated with RES$_P$ (Table 1). Overall DS$_P$ contribute $3.8\ Sv$ (13%) to the southward NADW transport at $53°N$, with about $3.4\ Sv$ (13%) within the density range of $27.62 \leq \sigma_0 < 27.86\ kg\ m^{-3}$ and $0.4\ Sv$ (11%) at densities $\sigma_0 \geq 27.86\ kg\ m^{-3}$. ISR$_P$ amount to $1.9\ Sv$ (6%) throughout the NADW $\sigma_0$ range, with $1.7\ Sv$ (6%) for the density range $27.62 \leq \sigma_0 < 27.86\ kg\ m^{-3}$ and $0.2\ Sv$ (5%) at densities $\sigma_0 \geq 27.86\ kg\ m^{-3}$ (Table 1).

The main spreading pathways from the respective sources to $53°N$ (Figure 3) are largely concentrated within the boundary currents for all four particle categories. In the north–eastern Labrador Sea, near Cape Desolation (west of Greenland), the pathways fan out for all categories, most probably related to the eddy activity here (Rieck et al., 2018), namely the shedding

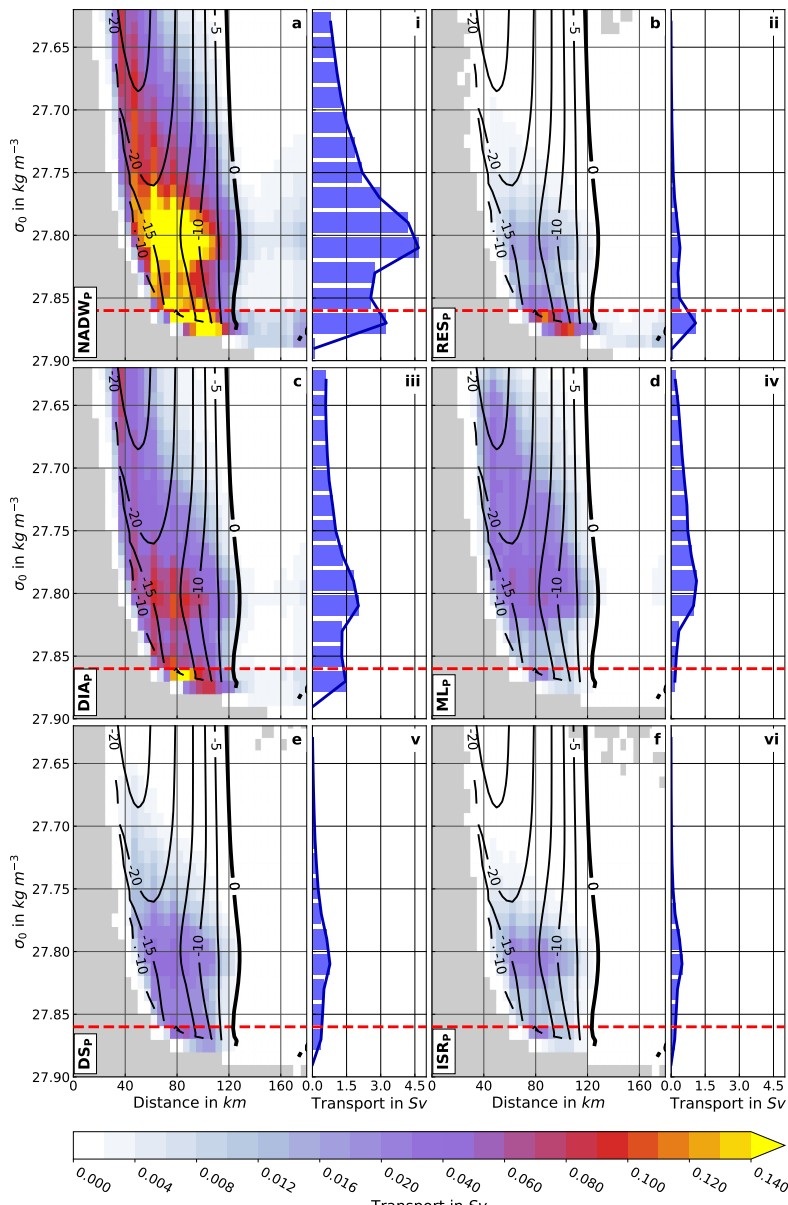

**Figure 2.** Mean transport distribution at $53°N$ in $Sv$ in density space (a to f, shading, $5\ km \times 0.01\ kg\ m^{-3}$ distance–density bins) and mean transport accumulated along $53°N$ in $Sv$ (i to vi), $0.02\ kg\ m^{-3}$ bins). The red dashed lines mark the mean $\sigma_0 = 27.86\ kg\ m^{-3}$ isopycnal. Black contours in a to f are mean meridional velocities in $cm\ s^{-1}$. Note the non–linear color scale ($0.002\ Sv$ intervals up to $0.02\ Sv$ and $0.01\ Sv$ intervals starting from $0.02\ Sv$) for a to f. Mean transport distributions are shown for (a/i) NADW$_P$, (b/ii) RES$_P$, (c/iii) DIA$_P$, (d/iv) ML$_P$, e/v DS$_P$ and (f/vi) ISR$_P$ (see section 2.1.3 for details of the definitions).

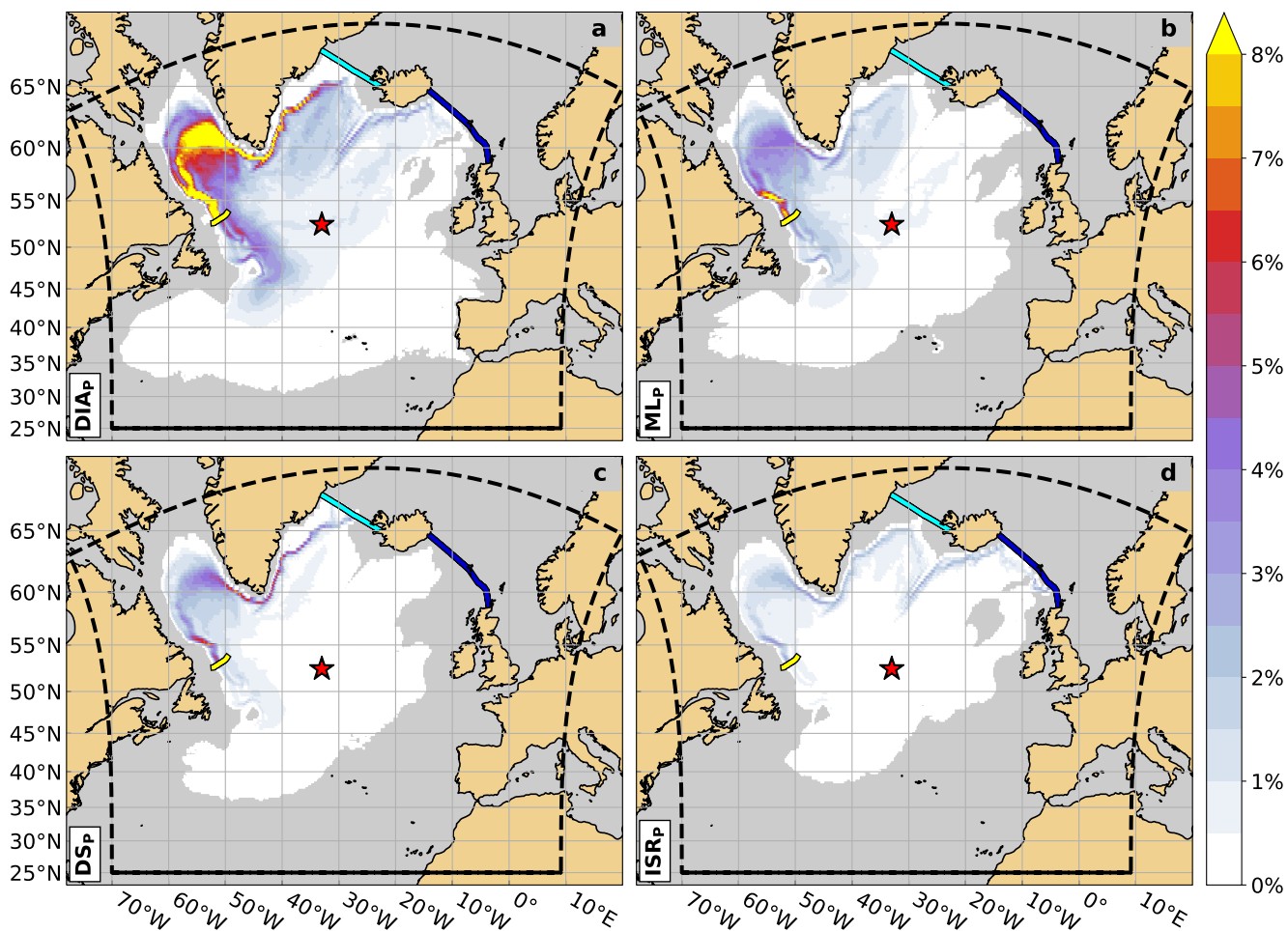

**Figure 3.** Particle pathways associated with most of the volume transport, calculated as transport–weighted probability density maps ($1/4° \times 1/4°$ bins) of the different particle categories: (a) $DIA_P$, (b) $ML_P$, (c) $DS_P$ and (d) $ISR_P$ (see section 2.1.3 for details of the definitions). The Charlie–Gibbs Fracture Zone is indicated by the red star. The yellow line marks the $53°N$ section, the light and dark blue lines mark the Denmark Strait and Iceland–Scotland Ridge sections, respectively. The black dashed line indicates the boundary of the experiment domain.

of Irminger rings. Furthermore, all particle categories exhibit pathways extending southward from $53°N$ (Figures 3). These pathways become more distinct on longer time scales and represent the recirculation south of $53°N$ at the Orphan Knoll region (not shown).

### 3.1.1 Diapycnal Mass Flux

About half the NADW transport at $53°N$, 14.3 $Sv$ (48%), is associated with diapycnal mass flux (Table 1). The majority of
255 $DIA_P$ enter the NADW density range within the boundary current in the Labrador Sea (5.5 $Sv$, Table 2) and Irminger Sea

**Table 2.** Mean transport contributions (2010-2019) from the Labrador Sea (LS) and Irminger Sea (IS), the Iceland basin, as well as the remaining SPNA south of $52°N$ (southern SPNA) in $Sv$. The regions are outlined by the blue dash–dotted lines in Figure 1. The Labrador and Irminger Sea contributions are separated into an interior and boundary component (water depths shallower than $3,000\ m$ in the Labrador Sea and $2,000\ m$ elsewhere). Transport contributions are given for DIA$_P$ and ML$_P$ (see section 2.1.3 for details of the definitions). Values are given for $\sigma_0 \geq 27.62\ kg\ m^{-3}$ (NADW) and $27.62 \leq \sigma_0 < 27.86\ kg\ m^{-3}$ (LSW), the difference between the two corresponds to the transport associated with $\sigma_0 \geq 27.86\ kg\ m^{-3}$ (lNADW).

| | | LS | | IS | | Iceland | southern |
|---|---|---|---|---|---|---|---|
| | | interior | boundary | interior | boundary | basin | SPNA |
| DIA$_P$ | $\sigma_0 \geq 27.62\ kg\ m^{-3}$ (NADW) | $0.6\ Sv$ | $5.5\ Sv$ | $0.5\ Sv$ | $4.2\ Sv$ | $2.4\ Sv$ | $1.1\ Sv$ |
| | $27.62 \leq \sigma_0 < 27.86\ kg\ m^{-3}$ (LSW) | $0.6\ Sv$ | $5.0\ Sv$ | $0.4\ Sv$ | $3.8\ Sv$ | $2.2\ Sv$ | $0.8\ Sv$ |
| ML$_P$ | $\sigma_0 \geq 27.62\ kg\ m^{-3}$ (NADW) | $2.6\ Sv$ | $3.3\ Sv$ | $0.5\ Sv$ | $0.6\ Sv$ | $< 0.1\ Sv$ | $0.2\ Sv$ |
| | $27.62 \leq \sigma_0 < 27.86\ kg\ m^{-3}$ (LSW) | $2.5\ Sv$ | $3.3\ Sv$ | $0.5\ Sv$ | $0.5\ Sv$ | $< 0.1\ Sv$ | $0.2\ Sv$ |

($4.6\ Sv$, Table 2) at depths between $600$ and $1,100\ m$ (not shown). Only a very small portion is added in the basin interiors (Figure 4 a, Table 2) at depths below $1,300\ m$ (not shown). The Iceland basin, adding $2.4\ Sv$, and the southern SPNA, adding $1.1\ Sv$, consequently play only a small role for the total NADW with DIA$_P$ origin. After their densification, particles of this category spread (Figure 3 a), additionally to the boundary currents, throughout the basin interior of the western SPNA with further pathways from the Iceland basin following the $1,000\ m$ isobath (Figures 3 a and 1) along the Reykjanes Ridge and through the CGFZ. Most particles pass through the central Labrador Sea before exiting it via the DWBC at $53°N$ (Figure 3 a). Most of the DIA$_P$ are formed at densities $27.62 \leq \sigma_0 < 27.86\ kg\ m^{-3}$, where the Labrador Sea exceeds the Irminger Sea by $1.4\ Sv$. Similar amounts of lNADW ($\sim 0.5\ Sv$) are added in both basins, with the Iceland basin and the southern SPNA again only playing a minor role with $2.2$ and $0.8\ Sv$, respectively (Table 2). The respective maxima of the mean transport are found just south of DS until $65°N$, east and west of Cape Farewell and between Hamilton Bank and $53°N$. Minor contributing regions are found along the continental slopes south of the ISR, the Reykjanes Ridge and within the Labrador Sea (Figure 4 a) most probably following the eddy track of Irminger Rings shed at Cape Desolation (Prater, 2002; Hátún et al., 2007; Rieck et al., 2018). Contrary to the boundary current the interior does not show as elevated values and a spread over a larger area (Figure 3 a). These patterns of the total NADW are similarly found for the density ranges $27.62 \leq \sigma_0 < 27.86\ kg\ m^{-3}$ and $\sigma_0 \geq 27.86\ kg\ m^{-3}$ (not shown). As the particle sources are located within or nearby boundary currents, which exhibit strong velocities, the particles can have rather short transit times. The transit time distribution peaks between 0 and 1 years, with more than a third of the volume transport associated with DIA$_P$ reaching $53°N$ within this time span (Figure 5 a). After approximately 1.94 years $50\%$ of the transport associated with DIA$_P$ has passed $53°N$ (Figure 5 a).

### 3.1.2 Mixed Layer

The second largest supplier of NADW, with $7.2\ Sv$ ($24\%$) of the $30.1\ Sv$ of NADW passing $53°N$, was found to originate from the mixed layer (ML$_P$, Table 1). The particles leave the mixed layer between November and June with a peak export between

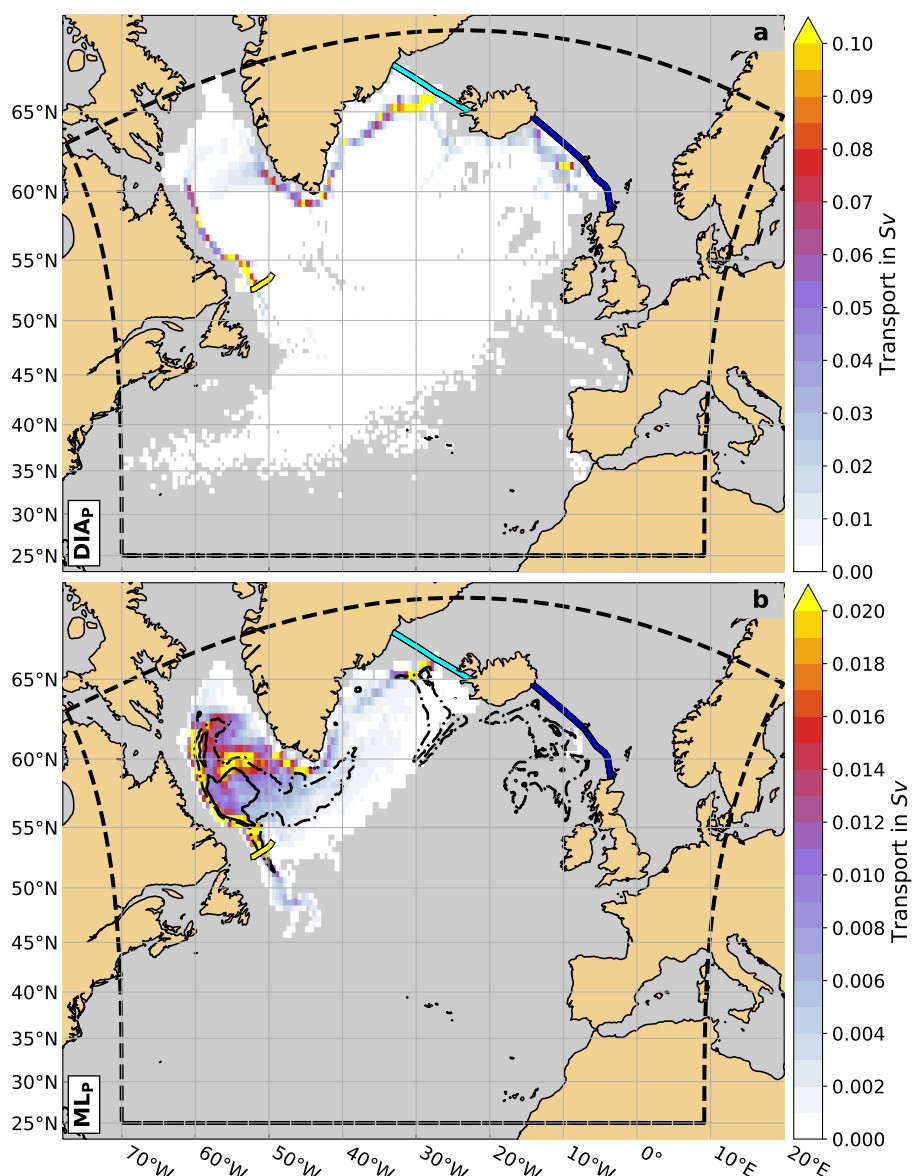

**Figure 4.** Locations of origin, calculated as mean transport in $Sv$ (shading, $1/2° \times 1/2°$ bins) for (a) $DIA_P$ and (b) $ML_P$ (see section 2.1.3 for details of the definitions). In (b), the black solid contour marks the 2000-2019 mean DJFM mixed layer depth of $500\,m$. The black dash–dotted contour marks the 2000-2019 mean of the annual maximum mixed layer depth of $500\,m$. The period 2000-2019 is chosen to capture the period where the vast majority of $ML_P$ are circulating. The yellow line marks the $53°N$ section, the light and dark blue lines mark the Denmark Strait and Iceland–Scotland Ridge sections, respectively. The black dashed line indicates the boundary of the experiment domain.

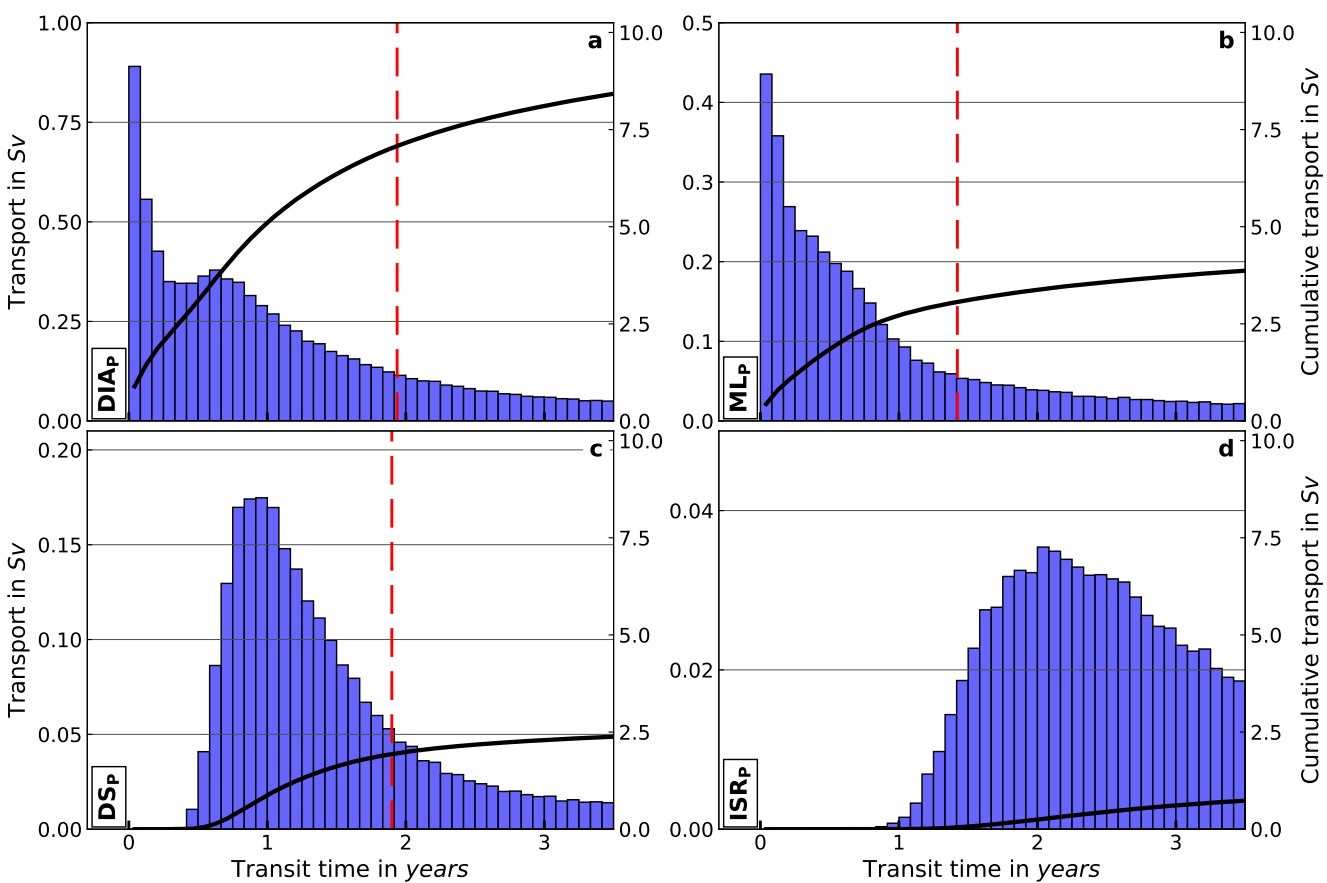

**Figure 5.** Transit time distributions, calculated as mean volume transport in $Sv$ (bars, lefthand scale) and cumulative mean volume transport in $Sv$ (black lines, righthand scale) as a function of transit time (1 $month$ bins) for (a) DIA$_P$, (b) ML$_P$, (c) DS$_P$ and (d) ISR$_P$ (see section 2.1.3 for details of the definitions). The red dashed lines mark the time after which half the transport associated with the respective category has reached $53°N$. Note the different $y$–axis scales for the bar plots.

February and April (not shown) dominantly within the central Labrador Sea or the boundary current along the Labrador shelf break (Figure 4 b). Elevated values can also be found south of DS and west of the Faroe Islands, which could possibly be related to the shallow pathways of the particles at these locations. There are minor, but noticeable contributions from the
Irminger Sea and south–west of Cape Farewell. These particles tend to leave the mixed layer at shallower depths and lower densities compared to particles from the Labrador Sea (Figure A1). The Labrador Sea, however, dominates as source region of this particle category with $5.9\ Sv$ (82% of the total ML$_P$ transport) compared to $1.0\ Sv$ (14% of the total ML$_P$ transport) from the Irminger Sea (Table 2). Within the Labrador Sea, the contribution from the boundary regions dominates with $3.3\ Sv$ over the interior contribution with $2.6\ Sv$ (Table 2). Half of the volume transport associated with ML$_P$ reaches $53°N$ within 1.42
285    years (Figure 5 b) which we expected due to the close proximity of the source regions to $53°N$.

### 3.1.3 GSR

With $5.7\ Sv$ ($19\%$) of the NADW transport at $53°N$, the water passing DS (DS$_P$, $3.8\ Sv$ or $13\%$) and the ISR (ISR$_P$, $1.9\ Sv$ or $6\%$) are the third biggest source of supply for NADW passing $53°N$ (Table 1). These volume transports are comparable to previous model analyses (DS $3.1 \pm 0.4\ Sv$ and ISR $1.3 \pm 0.2\ Sv$; Biastoch et al. (2021)). These values compare well with the transport estimates from observations, ranging from $3.1\ Sv$ (Jochumsen et al., 2017) to $3.5\ Sv$ (Harden et al., 2016) at DS and are slightly lower than the observed $2.2\ Sv$ (Hansen et al., 2016; Rossby et al., 2018) to $2.7\ Sv$ (Berx et al., 2013) at the ISR.

Particles that pass through DS within the NADW density range populate majorly the two $\sim 600\ m$ deep troughs of the strait, with domination of the deeper one just west of Iceland (Figure 3 c). Both pathways then merge south of DS and follow the East and West Greenland shelf break until reaching the Labrador Sea. DS$_P$ have a density between $27.70$ and $27.88\ kg\ m^{-3}$ (Figure 2 e, v). The longer the particles take to reach $53°N$, the more particles recirculate through the basin interiors of the Irminger and the Labrador Sea (not shown). The transit time distribution peaks between 1 and 2 years of advection (Figure 5 c), with $50\%$ of the associated transport arriving at $53°N$ within $1.90$ years.

NADW particles crossing the ISR are strongly concentrated within the boundary currents after entering the SPNA majorly through the Faroe Bank channel and a trough in the Iceland–Faroe Ridge just east of Iceland (Hvalbakshalli slope, Hjartarson et al., 2017, Figure 3 d). ISR$_P$ surround the Reykjanes Ridge between the $1,000$ and $2,000\ m$ isobaths (Figures 3 c and 1) and do not majorly pass through the CGFZ. The longer the particles take to reach $53°N$, the more particles tend to be advected through the basin interior of the SPNA (not shown). Due to the longer pathways of these particles, transit times tend to be longer. The transit time distribution peaks between 2 and 3 years (Figure 5 d), with $50\%$ of the associated transport arriving at $53°N$ within $4.78$ years. The decay with increasing transit times relative to the peak value is slowest for this category, as relatively more particles tend to be advected through the basin interior compared to the boundary currents.

Assuming that the shortest transit times are associated with the shortest distances traveled by the particles within a specific particle category, the average velocity a particle must at least have to reach $53°N$ can be calculated. This velocity is estimated to be $\sim 19\ cm\ s^{-1}$ for ISR$_P$ and $\sim 25\ cm\ s^{-1}$ for DS$_P$. Similar values are found for particles from the mixed layer and associated with diapycnal mass flux originating from areas close to the GSR. These values seem reasonable given the fact that mean velocities can easily exceed $20\ cm\ s^{-1}$ and reach more than $50\ cm\ s^{-1}$ at various depth levels in VIKING20X (e.g Figure 7 in Biastoch et al. (2021)).

### 3.1.4 Residuum

The volume that is not attributable to any of the above defined sources of NADW after $40$ years of advection amounts to $2.9\ Sv$ ($10\%$, Table 1). This residuum (RES$_P$) can be separated into particles circulating within the experiment domain for $40$ years and particles entering the domain across the southern boundary ($25°N$) or through Davis Strait. Particles circulating for $40$ years within the domain contribute $2.1\ Sv$ ($72\%$ of the total RES$_P$ transport). Particles entering from the south contribute $0.8\ Sv$ ($28\%$ of the total RES$_P$ transport) and particles passing through Davis Strait contribute $< 0.1\ Sv$ ($< 2\%$ of the total RES$_P$ transport) (not shown). The majority of RES$_P$ recirculate in the basin interior of the SPNA (Figure A2). The pathways

**Table 3.** Mean water mass property modifications of the four particle categories DIA$_P$, ML$_P$, DS$_P$ and ISR$_P$ (see section 2.1.3 for details of the definitions). Listed are potential density (referenced to 0 $dbar$, $\sigma_0$), absolute salinity ($S_A$), conservative temperature ($\Theta$) and the transformed volume in $Sv$ from source to $53°N$ (target section). When values are presented with "/", two major classes of properties are persistent for this particle category.

| | property | source | $53°N$ | transformed volume |
|---|---|---|---|---|
| DIA$_P$ | $S_A$ | $35.2\ g\ kg^{-1}$ | $35.16\ g\ kg^{-1}$ | $8.0\ Sv$ |
| ($14.3\ Sv$) | $\Theta$ | $5.4°C$ | $3.8°C$ | $9.3\ Sv$ |
| | $\sigma_0$ | $27.65\ kg\ m^{-3}$ | $27.80\ kg\ m^{-3}/27.85\ kg\ m^{-3}$ | $8.9\ Sv$ |
| ML$_P$ | $S_A$ | $35.14\ g\ kg^{-1}$ | $35.16\ g\ kg^{-1}$ | $1.9\ Sv$ |
| ($7.2\ Sv$) | $\Theta$ | $4.0°C$ | $3.8°C$ | $1.2\ Sv$ |
| | $\sigma_0$ | $\sim 27.75\ kg\ m^{-3}$ | $27.78\ kg\ m^{-3}$ | $1.7\ Sv$ |
| DS$_P$ | $S_A$ | $35.09\ g\ kg^{-1}$ | $35.16\ g\ kg^{-1}/35.19\ g\ kg^{-1}$ | $2.6\ Sv$ |
| ($3.8\ Sv$) | $\Theta$ | $0.8°C/2.8°C$ | $3.7°C$ | $3.1\ Sv$ |
| | $\sigma_0$ | $27.98\ g\ kg^{-1}/27.87\ kg\ m^{-3}$ | $27.80\ kg\ m^{-3}/27.87\ kg\ m^{-3}$ | $2.9\ Sv$ |
| ISR$_P$ | $S_A$ | $35.08\ g\ kg^{-1}$ | $35.16\ g\ kg^{-1}/35.19\ g\ kg^{-1}$ | $1.4\ Sv$ |
| ($1.9\ Sv$) | $\Theta$ | $0.2°C$ | $3.7°C$ | $1.6\ Sv$ |
| | $\sigma_0$ | $28.05\ kg\ m^{-3}$ | $27.80\ kg\ m^{-3}$ | $1.5\ Sv$ |

of the denser particles are mostly situated west of the Mid–Atlantic Ridge, while particles with lower densities are advected throughout the whole SPNA. This pattern emerges once the analysis is done in $\sigma_2$ (not shown). Most particles that cross the Mid–Atlantic Ridge pass through the CGFZ.

## 3.2 Water Mass Transformations

We evaluate the changes the water parcels undergo during their spreading routes from their point of origin to the $53°N$ target section. The evaluation is done for each particle class, except RES$_P$, ordered by the relative contribution of the respective particle class to the transport at $53°N$. All particle categories, apart from RES$_P$, show similar water mass property signatures at $53°N$. Hence, depending on the properties of absolute salinity ($S_A$), conservative temperature ($\Theta$) and density ($\sigma_0$) at the point of origin the water parcels undergo dissimilar changes along their pathways. Particles of DIA$_P$ and ML$_P$ origin densify through cooling, accompanied by freshening during spreading (Figure A3), whereas DS$_P$ and ISR$_P$ lighten through warming, accompanied by salinification from source to target (Figure A5). Volume–wise, the water mass transformations are more pronounced for DIA$_P$ and DS$_P$ than for ML$_P$ and ISR$_P$ (Table 3).

### 3.2.1 DIA$_P$

The water parcels associated with diapycnal mass flux undergo a significant cooling of $\Delta\Theta = -1.6°C$, and freshening of $\Delta S_A = -0.04\ g\ kg^{-1}$ along their pathways towards $53°N$. The freshening value arises from the salinity signature of DIA$_P$ at origin and target (Table 3 and Figure A3).

Freshening occurs mostly along the western flank of the Reykjanes Ridge, along the continental slope around Greenland and off Labrador, as well as in the interior Labrador Sea (Figure A4 a). East of Cape Farewell, the freshening occurs at depths around $1,300$ to $1,500\ m$. South of Cape Farewell and Cape Desolation values reach up to $\sim 2,000\ m$, while within the Labrador Sea the freshening occurs mostly shallower than $1,000\ m$. South of $53°N$ the transformation can occur at depths deeper than $2,000\ m$, however, these transformations are less important in terms of volume (Figure A4 a, c). Locations and 340 depths of the cooling are similar to the freshening (Figures A4 b, d), however, volume–wise the cooling is less pronounced along the eastern flank of the Reykjanes Ridge compared to the freshening (Figure A4 a-b).

    The cooling dominates over the freshening leading to a density increase (from $\sigma_0 = 27.65\ kg\ m^{-3}$ to $\sigma_0 = 28.05\ kg\ m^{-3}$) and narrows the property ranges of all three variables at $53°N$ compared to the source regions. Due to the nature of the diapycnal mass flux the intensive change of particle properties along the spreading pathways between source and target region 345 is not surprising. As described in section 3.1, most DIA$_P$ particles originate from the shelf breaks around Greenland and the Labrador Sea and get advected along the boundary current. Here, the elevated property exchange with the ambient shelf water is leading to freshening. Additionally, cooling can occur here within the NADW property range due to air-sea exchange.

### 3.2.2 ML$_P$

At their source regions, particles from the mixed layer are cooler and fresher than DIA$_P$ at their respective origin (Table 350 3). In this experiment particles from the mixed layer majorly originate from the Labrador Sea with slight domination of the boundary regime ($3.3\ Sv$) over the basin interior ($2.6\ Sv$). A small volume also originates from the southeast of the section and reaches $53°N$ after recirculation at Orphan Knoll (Figure 4 b). Only a relatively small amount of particles originate from the regions where the deepest mixed layers occur (black solid line in Figure 4 b). Due to the definition of the origin of ML$_P$, which is associated with the point where a particle exits the mixed layer within the NADW density range, this is 355 not surprising. The property transition occurs majorly in the marked mixed layer (Figure 4 b) and the water parcels are then advected to the associated point of origin, which is outside of the marked area. Hence, the origin locations only partially coincide with regions where deep mixed layers potentially occur (black dash–dotted line in Figure 4 b). ML$_P$ show a smaller value range compared to DIA$_P$ for all three parameters (Figure A3). Even though no distinct peak is discernible in the ML$_P$ $\sigma_0$ signature at their origin ($\Theta = 4.0°C$, $S_A = 35.14\ g\ kg^{-1}$), a slight cooling and salinity increase is apparent, leading to a slight 360 densification ($\Theta = 3.8°C$, $S_A = 35.16\ g\ kg^{-1}$ and $\sigma_0 = 27.78\ kg\ m^{-3}$ (Table 3 and Figure A3). These properties at origin and $53°N$ compare well with literature values for LSW (Liu and Tanhua, 2021). ML$_P$ undergo the least volumetric transformation of the presented particle categories. This is not surprising, as ML$_P$ are densified through convection and then, once cut off from the atmosphere, advected majorly adiabatically along isopycnals.

### 3.2.3 GSR

In contrast to the particles of the DIA$_P$ and ML$_P$ categories, the DS$_P$ and ISR$_P$ spread predominantly along the boundary currents. As mentioned before, DS$_P$ feature a mixture of different water types with similar salinity ($S_A = 35.09 \ g \ kg^{-1}$) and varying temperature signature (close to Greenland shelf: $\Theta = 0.8°C$; close to Iceland: $\Theta = 2.8°C$) within the NADW, but both undergo similar transitions south of DS (Table 3). DS$_P$ and ISR$_P$ both feature a decrease in density due to similar property transitions along their spreading pathways, as they undergo a substantial salinity increase and warming (Table 3 and Figure

A5). For the decrease in density, the temperature increase dominates over the increase in salinity. The salinity increase (to the point where $50\%$ of the total increase is reached) occurs for both particle categories just after crossing the GSR; close to the East Greenland shelf break just south of DS (DS$_P$) and along the ISR slope between Iceland and the Faroe Islands (ISR$_P$) within the $1,000$ and $2,000 \ m$ isobaths (Figure A6 and 1) and is followed by a continuous decrease in salinity until $53°N$ (Figure A7). The major salinity increase is reached at depths mostly shallower than $600 \ m$ for both categories and implies mixing with the

ambient upper AMOC water (Figure A6). Due to the shallowness of the overflows over the GSR, the mixing of the ISR$_P$ and DS$_P$ NADW water parcels with warmer and more saline upper AMOC waters just south of the overflows is not surprising. After this strong entrainment at rather shallow depths, the NADW spreads southward along isopycnals that increase their depth towards the south (Figure A8 c, d). Due to this relative sinking and the physical properties of the boundary current, some more diapycnal mixing (between $1,000$ and $1,500 \ m$), less intense than near the GSR, occurs along the east and west Greenland

slopes. Further enhanced mixing is found south and west of Cape Farewell around the Eirik Ridge at depths between $1,500$ and $2,000 \ m$, which results in further lightening of the DS$_P$ and ISR$_P$ within the NADW density range.

## 4   Discussion

In order to assess the mean relative contributions of the different sources of NADW passing the southern exit of the Labrador Sea at $53°N$, a Lagrangian particle experiment was conducted in the high-resolution ocean model VIKING20X-JRA-OMIP.

Each particle represents a defined volume and retains it along its trajectory, similar to stream tubes in a steady flow (van Sebille et al., 2017). Since the volume of each particle is preserved, but its properties are allowed to change along its trajectory, this resulted in the evaluation of the various sources, pathways, transit time scales and property transitions that NADW water parcels are subject to during their spreading from their origin in the SPNA to $53°N$.

### 4.1   Origins of NADW in observations vs. VIKING20x-JRA-OMIP

Concerning the volume transports of the respective particle classes, our results are not directly comparable to existing literature. Usually, the transports at $53°N$ are classified after their water mass properties into LSW, NEADW and DSOW which are of course related to their origins and have defined properties in temperature, salinity, density and/or potential vorticity (Zantopp et al., 2017; Liu and Tanhua, 2021). Observations of Zantopp et al. (2017) find a relative contribution of 50:50 of LSW ($14.9 \pm 3.9$ Sv) and lNADW ($15.3 \pm 3.8$ Sv) to the $30 \ Sv$ NADW transport at $53°N$. In contrast to this we find, using the previously

defined water mass definitions in the model (Handmann et al., 2018), only a very small amount of the modelled Eulerian NADW water transport of $30.1\ Sv$ at $53°N$ associated with lNADW ($3.4\ Sv$, Table 1). In order to evaluate the respective origin of the total NADW transport at $53°N$ in the model without being biased towards any predefined density interval we did not use rigid density classes in the following analyses but classified the transport volumes after their specific formation origin. Our experiment reveals that the specific particle categories are not primarily linked to an overall density definition at $53°N$ but rather to a similar formation region in combination with a specific transformation history. Just like the classical understanding of water masses, the densities are ordered with the densest components at origin at the ISR ($\sigma_0 = 28.05\ kg\ m^{-3}$) and DS ($\sigma_0 = 27.98\ kg\ m^{-3}$ and $\sigma_0 = 27.87\ kg\ m^{-3}$), and the lightest component from the mixed layer ($\sigma_0 = 27.75\ kg\ m^{-3}$) (Table 3). For the overflow component, a transport increase from the sills ($6\ Sv$ (Jochumsen et al., 2012; Hansen et al., 2010)) to the boundary current east of Greenland ($9\ Sv$ (Bacon et al., 2010)) to $53°N$ ($15\ Sv$ (Zantopp et al., 2017)) is observed. In observations the lNADW component is hence associated with the formation region in the Nordic Seas plus an added volume through entrainment of ambient water south of the overflow sills at the DS and ISR and through diapycnal mass flux into the respective density class in the SPNA through e.g. mesoscale eddies. Hence, these additional $\sim 10\ Sv$ are most probably represented in our analyses by the $DIA_P$ water class, even though they do not forcibly belong to the densest component in the model. Biastoch et al. (2021) show comparable Eulerian transports at the GSR between VIKING20X-JRA-OMIP and observations. This coincides with the total NADW sourced at DS ($3.8\ Sv$) found in this experiment (Table 1). As mentioned by Zou et al. (2020b) and Bower and Furey (2017) the water originating from the ISR can spread following very diffusive pathways. As we are only sampling water parcels passing $53°N$, the $1.9\ Sv$ we found to be originating from the ISR seem to be reasonable (Table 1). Consequently, in the model the dense overflow component from the GSR looses volume towards lighter densities laying within the predefined LSW density range. This is most probably caused by larger than observed entrainment of light water along the spreading pathway just south of the sill (Beismann and Barnier, 2004; Legg et al., 2009) and underlines a contrast to observations where the lNADW volume grows along the spreading route. On the other hand, in our experiment, the densest water below $\sigma_0 = 27.86\ kg\ m^{-3}$ cannot be assigned to a specific source after $40$ years of advection (Table 1). Most probably, this dense water is introduced at the initialization of the model and not refreshed or majorly changed and rather reduced towards lighter densities during the model run. Pathways of $RES_P$ are concentrated west of the Mid–Atlantic Ridge (Figure A2). This is related to the fact that the residuum mostly contains particles with very high densities that recirculate within the western SPNA (Figure 2 b, ii), and are unable to cross the Mid–Atlantic Ridge. To conclude, we find that the density interval with the major transports in NADW at $53°N$ around $\sigma_0 = 27.80\ kg\ m^{-3}$ is not only associated with one source. Instead multiple sources contribute with different relative importance to similar density regimes, though the $DIA_P$ and $ML_P$ dominate (Figure 2).

With $48\%$ of the total NADW and LSW transport, the $DIA_P$ represent the majority of NADW (LSW) at $53°N$ in this experiment (Table 1). This result aligns with the results of Lumpkin et al. (2008) and Marsh et al. (2005), who found that most of the LSW, leaving the SPNA southward, originates from subsurface diapycnal mixing, without contact to the atmosphere, rather than directly from the mixed layer as a result of air–sea fluxes.

## 4.2 Origin in the basin interior vs. the boundary current

The DIA$_P$ contributing to the NADW transport at $53°N$ are majorly confined to the continental shelf break (Figure 4 a, Table 2). Only small diapycnal mass flux is visible in the central Labrador Sea possibly due to mixing induced through eddies shed at Cape Desolation and even smaller, non significant numbers are found in the basin interior of the Irminger Sea or the Iceland basin (Figure 4 a, Table 2) (Prater, 2002; Hátún et al., 2007; Rieck et al., 2018). This pattern of densification along the buoyant boundary currents is shown in multiple idealised and realistic model studies (Spall, 2004; Xu et al., 2018; Katsman et al.,

2018; Brüggemann and Katsman, 2019; Georgiou et al., 2019; Sidorenko et al., 2020), as well as in observations (Waterhouse et al., 2014). Katsman et al. (2018) show that sinking of water masses occurs where friction plays an important role, thus close to the continental boundary. However, they only consider downwelling in depth space, thus the net sinking is not necessarily associated with a change in density. Based on an idealized model, Brüggemann and Katsman (2019) showed that densification can also be related to transport of water masses from lower to higher densities. In this case water masses are advected laterally

via mesoscale eddies into the boundary current across an isopycnal, leading to a change in density. The true causes of this pattern, however, need to be explored in more detail in order to elaborate a profound hypothesis, based on the model's abilities. Here, the densification is understood as a result of diapycnal volume fluxes and mixing induced by strong density gradients below the mixed layer in the vicinity of steep topographic slopes and a respective enhanced eddy activity (Spall, 2001; Radko and Marshall, 2004; MacKinnon et al., 2013; Zhang et al., 2019). Consequently, the relative contribution of the basin interiors

is negligible as we showed (Table 2). Additionally, the diapycnal volume fluxes could be further linked to air–sea heat fluxes upstream of the respective NADW origin region through outcropping of the respective isopycnal (Walin, 1982; Grist et al., 2009; Marsh, 2000; Desbruyères et al., 2019; Petit et al., 2020, 2021). The arising density compensated shifts in temperature and salinity in the subpolar mode water (SPMW) just above the NADW could then facilitate densification along the net sinking pathways of SPMW, though this analysis is beyond the scope of this paper.

## 4.3 Hydrographic transformations along spreading pathways

Another aspect of diapycnal mixing is featured in the property change of DS$_P$ and ISR$_P$, the warming and salinification, from the GSR to $53°N$ (Table 3, Figure A5). These water parcels spread below the main thermocline and gain buoyancy during their southward propagation as expected (MacKinnon et al., 2013). We showed that the major part of the density decrease, at least $50\%$ of the salinification and warming, occurs south of the GSR sills (Figure A6). Here, the mixing driving this

density transformation is elevated due to enhanced turbulence through the high velocities (exceeding $20\ cm\ s^{-1}$ reaching up to $50\ cm\ s^{-1}$) at the sills and the sloping topography (Figure 1, Rudels et al., 2002; Koszalka et al., 2013; Garabato et al., 2019). This is in agreement with the results of Fogelqvist et al. (2003) and Devana et al. (2021), who find a massive impact of upper ocean properties on the NADW passing the GSR channels due to high spill velocities enhancing shear instabilities towards the overlaying upper AMOC waters. South of DS, additionally to the upper AMOC waters, fresh and cold East Greenland Current

water is another possible ambient water mass to be entrained (Dickson and Brown, 1994). Hence, we conclude in concurrence with Jochumsen et al. (2015) and Devana et al. (2021) that changes in the mixing ratio and the respective water properties

of entrained waters can influence the downstream NADW properties originating from the GSR majorly. During the spreading along the boundaries a net sinking in depth space of the NADW from the GSR is found (Figure A8 c-d), in correspondence to Katsman et al. (2018). For DIA$_P$ we found a cooling and freshening between the source and target section (Table 3, Figure A3). Mixing with colder and fresher water from the basin interior could play a role here (Spall and Pickart, 2003). ML$_P$ only show small property alterations along their spreading pathways probably related to the spatial closeness to the $53°N$ target section.

## 4.4 Contribution from the Mixed layer

Consistent with previous studies, both observational and model–based (Pickart et al., 1997; Marshall and Schott, 1999; Pickart et al., 2002; Cuny et al., 2005; Brandt et al., 2007; MacGilchrist et al., 2020; Georgiou et al., 2021), those mixed layer (ML$_P$) origins contributing majorly to the $53°N$ transport are located within the central Labrador Sea and the Western Boundary Current region in the Labrador Sea (Figure 4 b and Table 2). The contribution from the boundary regions exceeds the direct contribution from the interior (Table 2). In agreement with Koelling et al. (2022) the export of these ML$_P$ at $53°N$ is between February and April and the transit times between formation and export are only a few months (Figure 5 b).

The experiment also shows a small but noticeable contribution from the Irminger Sea and from south–east of Cape Farewell (Table 2 and 1, Figure 4 b). Contributions from these regions are to be expected, since the Irminger Sea and the southern tip of Greenland have been established as additional sites of deep convection (Pickart et al., 2003; Våge et al., 2008; de Jong et al., 2012, 2018; Rühs et al., 2021), although the relative importance of each of them is still under debate. However, the Irminger Sea only plays a minor role in the presented experiment, providing only $1.0 \, Sv$ ($14\%$) of the total volume transport associated with the mixed layer, compared to $5.9 \, Sv$ ($82\%$) from the Labrador Sea (Table 2). The convection area along with the produced density and volume produced through convection in the Irminger Sea is comparable to the Labrador Sea in the period 2015-2018 (Rühs et al., 2021). Here, we analyze the period 2010-2019 which is not regarding the possible strong inter-annual variations in relative contribution of the two basins to the overall mixed layer contribution to the NADW at $53°N$. Hence, it is possible that the Irminger Sea contribution is underestimated in the second half of our experiment. Furthermore, in accordance to Le Bras et al. (2020) and Rühs et al. (2021), the shallower components of convective water masses from the Irminger Sea tend to be lighter compared to water masses formed within the Labrador Sea (Figure A1). Thus, it is possible that particles leaving the mixed layer in the Irminger Sea undergo further transformation along their pathways towards $53°N$. If these particles experience a reduction in density to values lower than $\sigma_{DW}$ they would add volume to the SPMW but not to the NADW and are not covered in our experiment. On the other hand, if the density is increased to values higher than $\sigma_{DW}$ again at a later point, these particles would be attributed to a different region or a different source entirely.

## 4.5 Pathways of NADW

Due to our experimental setup and as expected from literature (Kieke and Yashayaev, 2015; Palter et al., 2016; Fischer et al., 2018), NADW is majorly advected within the boundary currents close to the continental slope or the shelf break (Figure 3). Already west of the Eirik Ridge we noticed an enhanced divergence of the particle pathways, which coincides with trajectories from RAFOS floats of the OSNAP float program (Zou et al., 2021). Near Cape Desolation the pathways further diverge into

the Labrador Sea, becoming less confined (Figure 3) due to bifurcation and the shedding of Irminger Rings (Cuny et al., 2002; Prater, 2002; Hátún et al., 2007; Higginson et al., 2011; Palter et al., 2016; Rieck et al., 2018). Thus, water parcels are transported along more diverse pathways from Greenland across the Labrador Sea before joining into the more confined DWBC off Labrador. South of $53°N$ all particle categories feature a cyclonic recirculation cell around Orphan Knoll (Figure 3), which is in agreement with previous studies (Lavender et al., 2000; Xu et al., 2010). Hence, a slight NADW volume formation is also possible south of $53°N$ possibly due to horizontal mixing or ocean–atmosphere interaction. These waters can then recirculate to $53°N$ and the Labrador Sea.

Overall, at $53°N$ the total Labrador Sea contribution ($12.0\ Sv$) to the formation of NADW dominates over the Irminger Sea contribution ($5.7\ Sv$) for the evaluated experiment period (Table 2). This seems to be in contrast with recent observation–based studies. Lozier et al. (2019) state that OSNAP East dominates the AMOC in the SPNA, rather than OSNAP West. The study by Bower et al. (2009) shows that interior pathways are likely to be at least equally important for the export of NADW from the SPNA. The experiment presented here only takes into account the volume transport at $53°N$, i.e. the NADW export within the DWBC. Thus, here we do not represent the relative contribution of each basin to the total SPNA NADW southward export, which would reflect the AMOC. Additionally, we analyze the trajectories in bulk, which can lead to the impression of the Labrador Sea dominating in NADW formation. To investigate the changing relative points of origin over time the analysis that we have done here for the mean could be done for each seeding particle set but this is exceeding the scope of this paper.

## 5   Conclusion

In this study we show that multiple sources of NADW passing $53°N$ contribute to similar density regimes. The classical view of density classes being directly linked to a common formation region holds only partly within our experiment. We rather find that different origins in combination with transformation processes such as diapycnal mixing along the pathways lead to water mass properties that can be very similar at the southern exit of the Labrador Sea. We found that water passing the GSR ($DS_P$, $ISR_P$) within the NADW layer lightens within the NADW class through warming, which is accompanied by salinification, just south of the sills by mixing with ambient water. Contrary, NADW water originating from densification through diapycnal mixing ($DIA_P$) or contact with the mixed layer ($ML_P$) rather densifies after entering the NADW density class through cooling, which is accompanied by freshening. Due to our focus on the NADW transport within the DWBC at $53°N$, in our experiment the volume contribution from the Labrador Sea dominates over the rest of the subpolar North Atlantic. Since we analyzed our experiment in an averaging manner for the period 2010-2019 the inter-annual variability of the different sources is not discussed here. The relative importance of origin regions and transformation processes over time is hence a topic left for further analysis.

*Data availability.*   The trajectory data, analyzed in the current study are available from the corresponding authors on request.

*Author contributions.* PH and AB defined and guided the overall research problem and methodology. JF developed and performed the Lagrangian experiment and did the analyses and figures. JF and PH wrote the manuscript. All co-authors discussed the analyses and contributed to the text. The authors declare that they have no conflict of interest. All authors agree to be accountable for the content of the work.

*Competing interests.* The authors declare that the research was conducted in the absence of any commercial or financial relationships that could be construed as a potential conflict of interest.

*Acknowledgements.* We thank Dr. Willi Rath for the support during the experimental setup. We thank Dr. Franziska Schwarzkopf for running the Viking20x-JRA-OMIP model. This work received support by the Initiative and Networking Fund of the Helmholtz Association through the project "Advanced Earth System Modelling Capacity (ESM)"

# Appendix A: Supplementary Figures

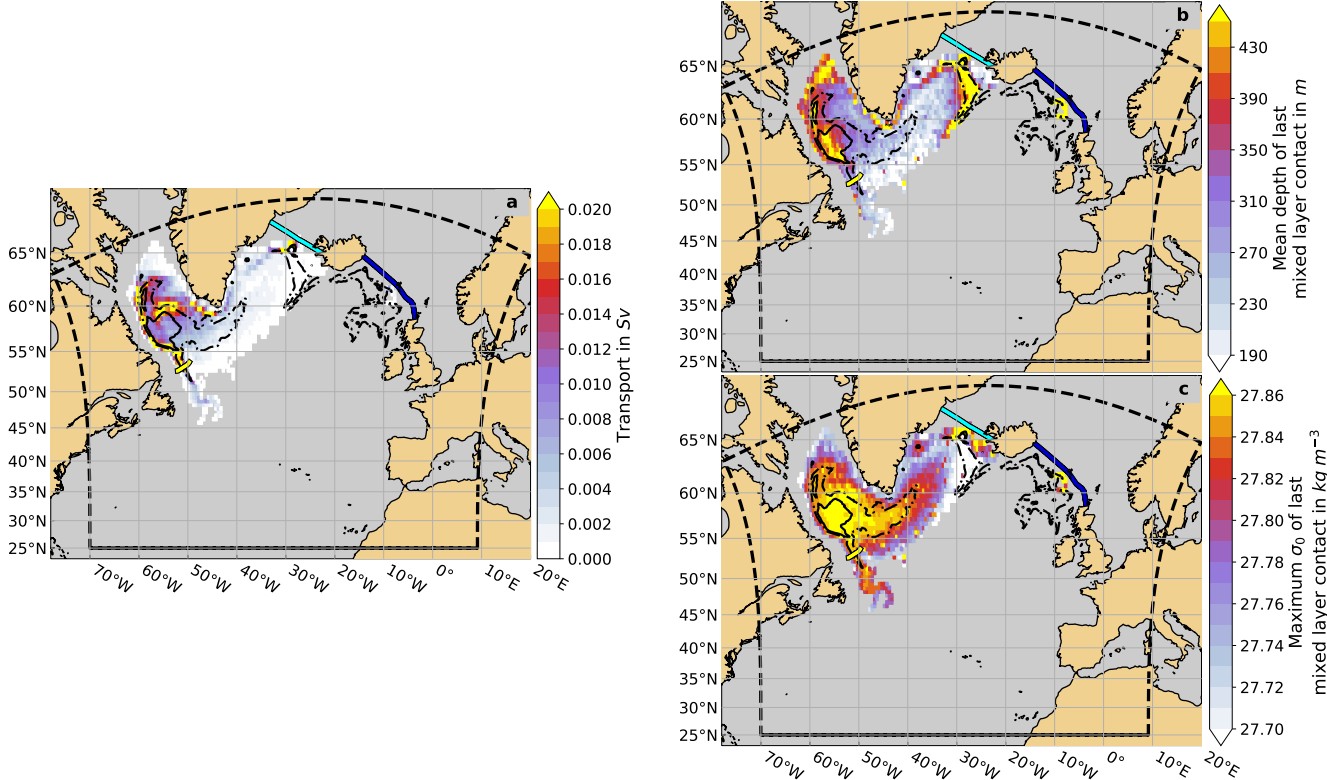

**Figure A1.** (a) Locations of origin, calculated as mean transport in $Sv$ (shading, $1/2° \times 1/2°$ bins) for $ML_P$ (see section 2.1.3 for details of the definitions). (b-c) Transport–weighted mean depth in $m$ (b) and maximum $\sigma_0$ in $kg\ m^{-3}$ (c) of last mixed layer contact per $1/2° \times 1/2°$ bin for $ML_P$. The black solid contour marks the 2000-2019 mean DJFM mixed layer depth of $500\ m$. The black dash–dotted contour marks the 2000-2019 mean of the annual maximum mixed layer depth of $500\ m$. The period 2000-2019 is chosen to capture the period where the vast majority of $ML_P$ are circulating. The yellow line marks the $53°N$ section, the light and dark blue lines mark the Denmark Strait and Iceland–Scotland Ridge sections, respectively. The black dashed line indicates the boundary of the experiment domain.

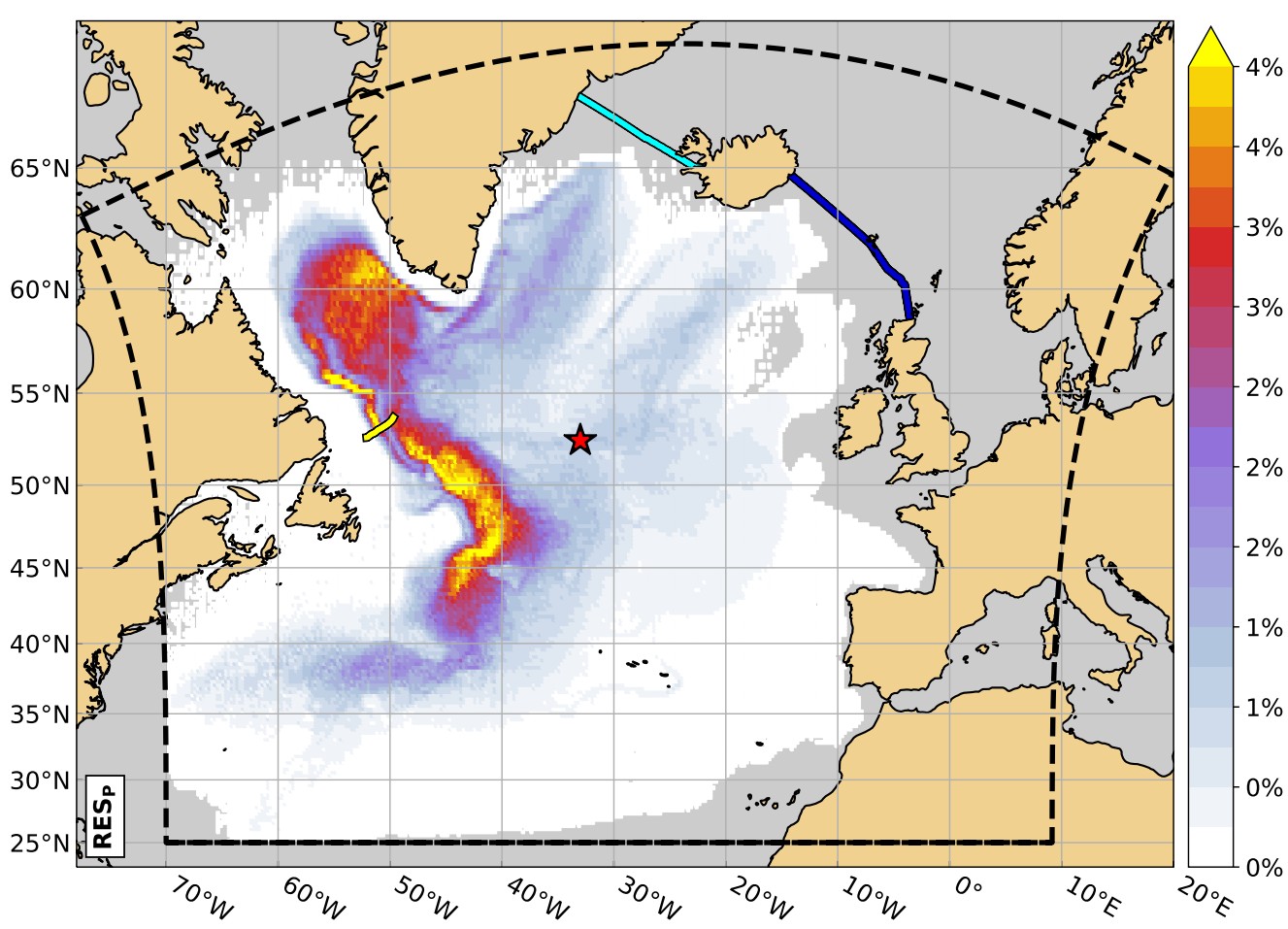

**Figure A2.** As in Figure 3, but for RES$_P$ (see section 2.1.3 for details of the definition).

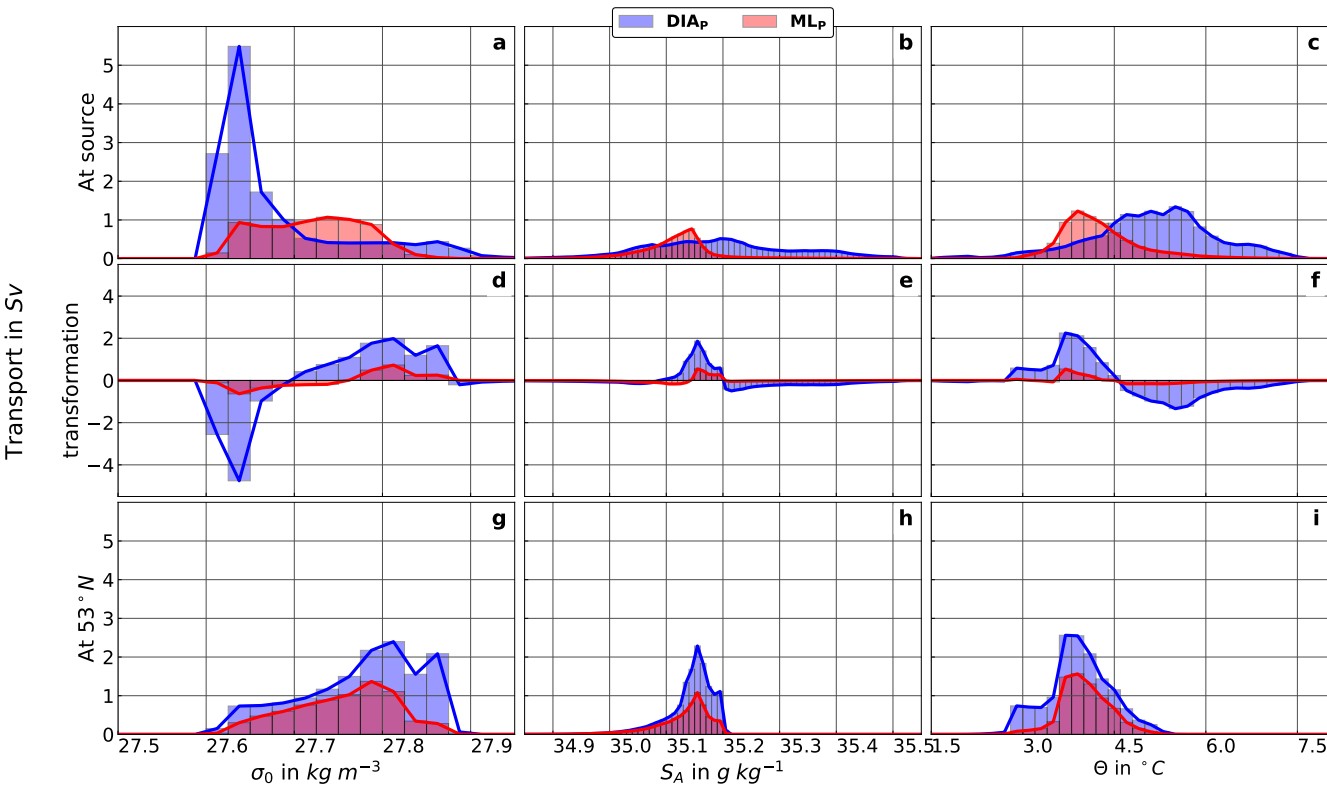

**Figure A3.** Mean water mass property modifications for DIA$_P$ (blue) and ML$_P$ (red) (see section 2.1.3 for details of the definitions). Shown are mean volume transport in $Sv$ per potential density (referenced to $0\ dbar$, $\sigma_0$, (a, d, g), $0.025\ kg\ m^{-3}$ bins), absolute salinity ($S_A$, (b, e, h), $0.01\ g\ kg^{-1}$ bins) and conservative temperature ($\Theta$, (c, f, i), $0.2°C$ bins) class at their source region (a to c) and at $53°N$ (g to i), as well as the volumetric property transformation (d to f).

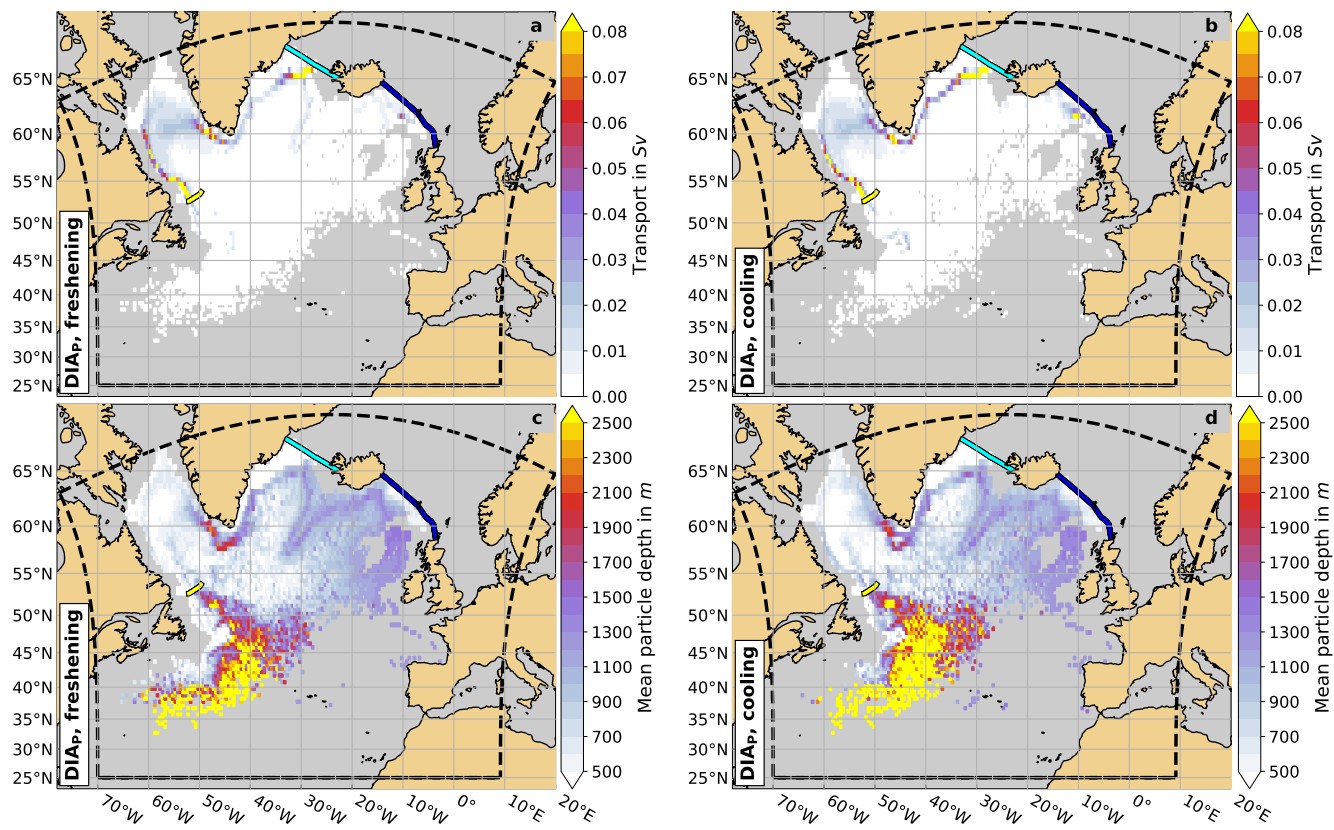

**Figure A4.** (a-b) Locations associated with most transport in $Sv$ and (c-d) mean particle depth in $m$ per $1/2° \times 1/2°$ bin for DIA$_P$. The particle locations are chosen as the locations where the difference in salinity between the particle's salinity minimum and its source is halved (a, c) and where the difference in temperature between the particle's temperature minimum and its source is halved (b, d). The yellow line marks the $53°N$ section, the light and dark blue lines mark the Denmark Strait and Iceland–Scotland Ridge sections, respectively. The black dashed line indicates the boundary of the experiment domain.

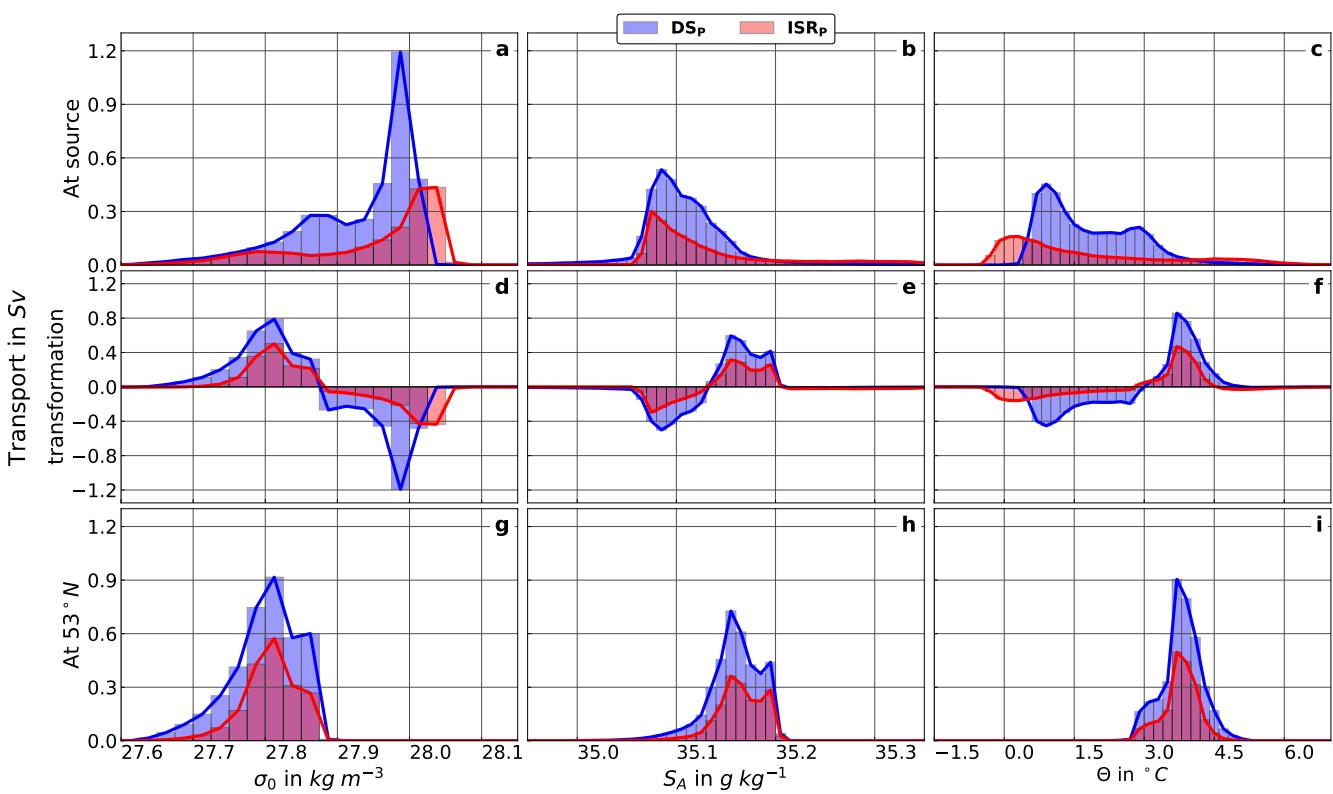

**Figure A5.** As in Figure A3, but for DS$_P$ (blue) and ISR$_P$ (red) (see section 2.1.3 for details of the definitions).

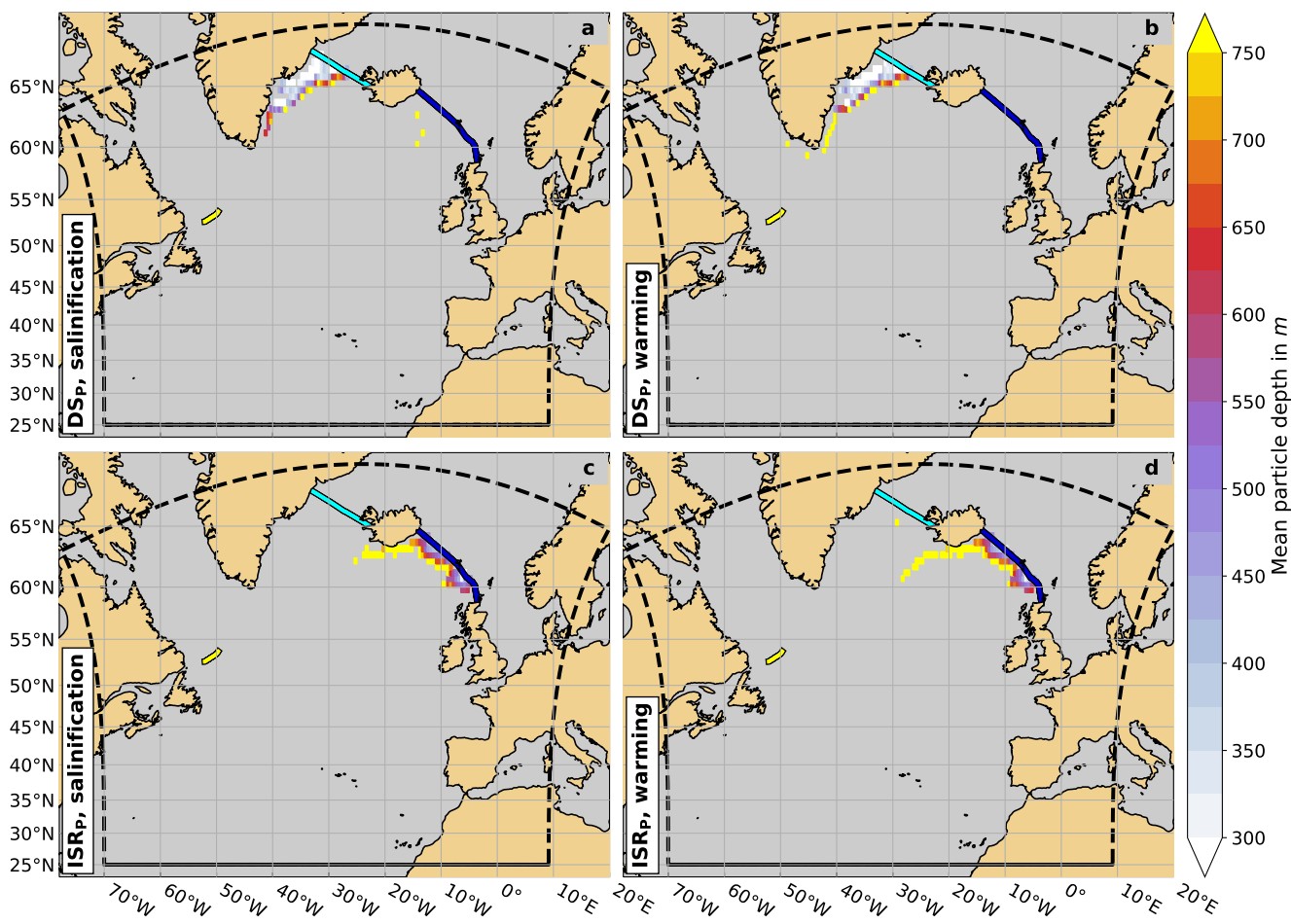

**Figure A6.** Mean depth of major (a, c) salinity and (b, d) temperature increase for DS$_P$ (a, b) and ISR$_P$ (c, d), calculated as the transport–weighted mean particle depth in $m$ per $1/2° \times 1/2°$ bin. The particle locations are chosen as the locations where the difference in salinity between the particle's salinity maximum and its source is halved (a, c) and where the difference in temperature between the particle's temperature maximum and its source is halved (b, d). The yellow line marks the $53°N$ section, the light and dark blue lines mark the Denmark Strait and Iceland–Scotland Ridge sections, respectively. The black dashed line indicates the boundary of the experiment domain.

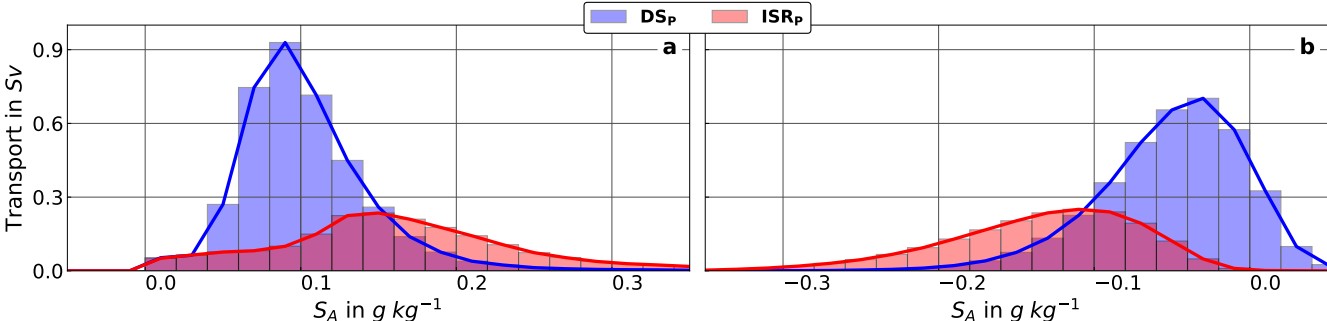

**Figure A7.** Difference in absolute salinity in $g\ kg^{-1}$ per $0.02\ g\ kg^{-1}$ bin for DS$_P$ (blue) and ISR$_P$ (red) (see section 2.1.3 for details of the definitions). Differences are calculated (a) between the location where the difference in salinity between a particle's salinity maximum and the source is halved and the source, i.e. the change in salinity from the source to the location where $50\%$ of the salinity increase occurs (see Figure A6 a, c). Furthermore, the differences are calculated (b) between particle release and the location where the difference in salinity between a particle's salinity maximum and the source is halved, i.e. the change of salinity from the location where $50\%$ of the salinity increase occurs (see Figure A6 a, c) to $53^{\circ}N$.

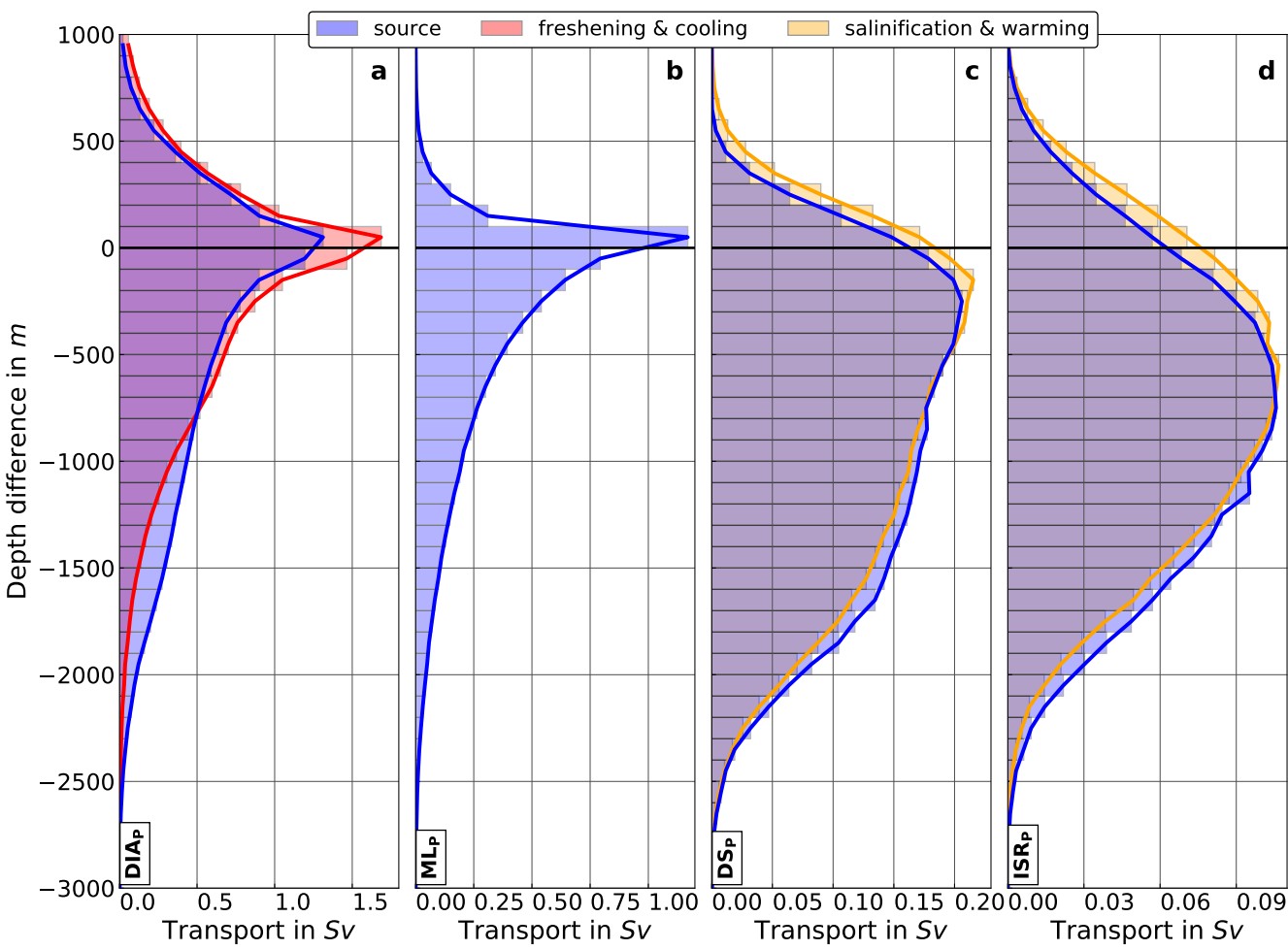

**Figure A8.** Evolution of particle depth between the particle source (blue) and $53°N$, as well as between the location where the major salinity and temperature decrease (red) or increase (orange) occurs and $53°N$ calculated as the mean transport per depth difference bin ($100\,m$) in $Sv$ for (a) $DIA_P$, (b) $ML_P$, (c) $DS_P$ and (d) $ISR_P$ (see section 2.1.3 for details of the definitions).

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
