# Peer review of "Major sources of North Atlantic Deep Water in the subpolar North Atlantic from Lagrangian analyses in an eddy-rich ocean model"

_EGUsphere, 2022_

## Referee Comment (RC1)

**Review on "Major sources of North Atlantic Deep Water in the subpolar North Atlantic from Lagrangian analyses in a high–resolution ocean model" by Fröhle et al.**

**General Comments**

The paper from Fröhle and colleagues investigates the relative contributions of the different sources of the North Atlantic Deep Water (NADW) that exits the Labrador Sea at $53^o$N. The manuscript outlines an analysis of Lagrangian particles in a high-resolution model to determine the NADW sources and its pathways and associated timescales from each source to the $53^o$N. The authors detail the interesting finding that within the subpolar North Atlantic the water mass transformation towards the density range of the NADW mainly happens through the process of diapycnal mixing (non-convective and convective).

These are interesting results that further our understanding of the high latitude ocean circulation, given all the present discussions of the Labrador Sea and its potential role, at different timescales, in the Atlantic Meridional Overturning Circulation (AMOC). Overall, the paper is generally well organized and clear, however some of the writing could be improved. I am pretty sure that this can be easily addressed by the authors and that might help to improve this already good paper. Thus, I recommend this paper for publication after major revision.

**Specific Comments**

1. Diapycnal mixing vs deep convection: Although the separation of the two processes might be trivial for most of the readers, I would suggest that the authors clarify a bit better the differences in the manuscript. I believe that the authors aim in distinguishing the diapycnal water mass transformations which are associated with deep convection, thus unstably stratified water masses, from water mass transformations, which are caused by internal diapycnal mixing across stratified density layers. Although in section 2.2 the authors clearly separate the two processes for the particles' categories, I would recommend a better explanation

of the two processes in the rest of the manuscript.

2. Guidance to the reader: I found myself wondering too many times at which figure/table should I look for many statements in the manuscript and in particular in section 3. I pointed out a few such examples below, but it is not an exhaustive list. Also, the terminology used in this manuscript regarding the source and the point of origin for each particle category is somehow mixed within the text.

3. Discussion (section 4): I feel that this section needs a better structure. It might be helpful to add some subsections to organize the discussion of your results. Many paragraphs are rather large with mixed information and difficult to follow. Furthermore, in many statements a reference to the relevant figure/table is missing. I would also suggest that the discussion begins with a short summary of the goal/methodology of this manuscript.

**Technical Comments**

1. **Title:** This study is based on the results of one experiment, right? Therefore, I suggest that you change ".. Lagrangian analyses.." to ".. Lagrangian analysis.." in the title.

2. **Line 1:** The North Atlantic Deep Water (NADW) ..

3. **Line 3:** ... components of NADW (namely..)

4. **Line 4 and elsewhere:** experiments –> experiment

5. **Line 5:** "according to the strength of the velocity field", this could be removed

6. **Line 5:** change "computed" to "traced"

7. **Line 6-7:** "Water masses were defined ... hydrography field", I don't see the importance of this sentence in the abstract. Consider removing this.

8. **Line 13:** Please consider rephrasing "... is hence dominated by the processes of diapycnal mixing and deep convection in the Labrador Sea."

9. **Line 15:** I believe that a short description of the AMOC would be beneficial.

10. **Line 16-18** Please rephrase this sentense in line with the general comment #1. Also, references of Straneo and Katsman are odd here. You could refer to Johnson et al. (2019)[1].

11. **Line 22:** .. of the North Atlantic Deep Water ..

12. **Line 26-28 & 31-32 & 33:** References are missing.

13. **Line 45:** further –> farther

14. **Line 46:** east Greenland –> East Greenland

15. **Line 53:** ".. is not finally understood.. " –> still remains unclear

16. **Line 54:** Consider adding more recent references than Lozier (2012) and Rhein et al. (2013).

17. **Line 60-62:** Please revise this sentence and add references of relevant studies.

18. **Line 68-69:** "Additionally, the observed .." –> Additionally, the deep convection in the Irminger Sea is more frequently evident in recent observational studies ... or similar.

19. **Line 75:** ".. and pathways of the single.." –> ".. and pathways of each of the deep water sources..

20. **Line 77:** section 5 is omitted here.

21. **Line 96:** .. along a section –> at a section

22. **Line 99-100:** Are the particles released over the whole water column in the vertical?

23. **Line 115:** How are the particles traced backward in time? Do you use the velocity/hydrography field averaged over a certain period and re-
* * *
[1]Johnson, H. L., Cessi, P., Marshall, D. P., Schloesser, F., & Spall, M. A. (2019). Recent contributions of theory to our understanding of the Atlantic Meridional Overturning Circulation. Journal of Geophysical Research: Oceans, 124, 5376– 5399

peat this as input for Parcels or do you use the daily snapshots from the release day of each particle and backwards?

24. **Line 142-144:** Is the inspection of the location of the particle's density change (within or outside the mixed layer) based on a time-mean value of the mixed layer depth?

25. **Line 155:** Why did you select the 3000 m isobath for the separation between the boundary current and the interior? Please also consider mentioning the reasoning behind this separation.

26. **Line 177 and elsewhere:** In my understanding, the point of origin is relevant only for the $DIA_p$ and $ML_p$ particles. If so, please mention this here and keep a consistency between the terms "source" and "point of origin" in the remaining of the manuscript.

27. **Line 78-79:** ".. based on their respective starting points." What do you mean here?

28. **Line 180:** "To determine .. respective basin." –> Rephrase to something like: "To determine the transport contribution of the different basins within the subpolar North Atlantic, we define ...". You could omit the limits of each area that you define and just refer to figure 2 for the definition of the different areas.

29. **Line 185-186:** You define the travel time of each particle based on the point of origin (point where the particle changes its density). However, in Figure 5 you include the transit time of $DS_p$ and $ISR_p$. Please revise this sentence.

30. **Figure 2:** Why did you choose to average the mixed layer depth over the 2000-2019 period and not over 1958-2019 or 2010-2019, which are periods that are discussed earlier in the manuscript? Also, this figure is mainly discussed after Figure 3. Consider changing the order of these figures. If you do so, you could remove the blue dashed lines that you use to define the areas in Figure 2 and add these lines in Figure 1.

31. **Table 1:** Please check the values. The transports/contributions associated with each particle category do not sum up to the total transport (i.e., for NADW: 101% instead of 100%, for LSW: 12.8+7.0+3.4+1.7+1.7 =

26.6 Sv instead of 27.7 Sv). There is also an extra parenthesis at 3 column, 4 row.

32. **Line 200:** I am missing a short introductory paragraph/sentences here. It is a rather rough beginning for the reader. It is not clear in which figure the reader should look.

33. **Line 283:** "The upper transport.." –> The lighter transport peak ..?

34. **Line 249:** "... is dominated by diapycnal mass flux and the particle residuum." –> "... is dominated by the $DIA_p$ and $RES_p$ particles.

35. **Line 207:** ($ML_p$, 7.0 Sv or 26%, Table 1)

36. **Line 214:** I don't feel this has been fully demonstrated. Perhaps adding the region of high EKE in one of the figures would help.

37. **Line 219:** "... within the boundary current in the Labrador Sea (5.5 Sv, **Table 2**) and Irminger Sea (4.6 Sv, **Table 2**) **at** depths between ..."

38. **Line 222-224:** I guess that this statement refers to Figure 4 a. However, I don't see the 1000 m isobath.

39. **Line 225-229:** References to the associated figure/table are missing. Also, consider adding the values of INADW in Tabel 2.

40. **Line 231:**"... single regions in the interior..". What do you mean?

41. **Line 240:** "Boundary Current" –> boundary current

42. **Line 248:** add a reference to Figure 5b.

43. **Line 253:** "Transport" –> transport

44. **Figure 5:** Consider making a new figure for the panels (e-f). You only mention these panels shortly in section 4 (Discussion).

45. **Line 259 and Figure 5:** The definition of the transit/travel time here differs from $DIA_p$ and $ML_p$, right?

46. **Line 267-268:** Please be more concrete here. What do you mean by "due to the importance of interior pathways"?

47. **Line 275-284:** Why not including a figure showing the major pathways of $RES_p$?

48. **Line 286-289:** Please revise; refer to earlier figure/table to support your statement and guide the reader of what is following in this section.

49. **300-301:** Do you mean the region south of $53^oN$? Is this still considered the interior of the SPNA?

50. **Line 312-319:** Here, many elements of discussion have already been discussed in section 3.1.2. Consider revising the text.

51. **Line 335:** ".. is followed by a continuous decrease in salinity until $53^oN$." Is this statement verified by a figure?

52. **Figure A2:** There are some extra parenthesis in the caption.

53. **Figure A3 and Figure A4:** Consider adding a title in each panel. Also, mention in the caption of figure A3 that this figure is for the DIA particles.

54. **Line 345-346:** Please revise this sentence.

55. **Line 357-358:** If the water masses are laterally advected within an iso-pycnal, how a change in density is then possible?

56. **Line 378-380:** Figure 5(e-f) is only shortly discussed here. I don't see the relevance here. Is the signal of downwelling only related to the particles that originate at the GSR? What about the other particle categories?

57. **Line 394-396:** Please revise.

58. **Line 405-406:** I guess that here you are referring only to the contribution of $ML_p$. If so, why? What about $DIA_p$?

59. **Line 412-420:** Please add references to the associated figures.

60. **Line 420-421:** Please revise.

61. **Line 422:** Volume transports of what?

62. **Line 438 and elsewhere:** stemming –> originating or similar.

---

## Referee Comment (RC2)

Review on "Major sources of North Atlantic Deep Water in the subpolar North Atlantic from Lagrangian analyses in a high-resolution ocean model" by Fröhle et al.

**Summary**

Fröhle et al. investigated the major sources of the North Atlantic Deep Water transports at the southern exit of the Labrador Sea using Lagrangian particle experiments in a high-resolution ocean model. They quantified contributions from different processes, including diapycnal fluxes (in and out of the mixed layer) and overflow from the Greenland-Iceland-Scotland ridge, to the total deep water transports. For each source, the associated pathways and transit times were discussed.

Overall, I find results from the study quite interesting as they show, in a model, what the subpolar deep western boundary current is composed of from a Lagrangian perspective. The manuscript was overall clearly written and the particle experiments were reasonably designed, supporting the major conclusions. Therefore, I recommend this manuscript for publication after addressing the following comments.

**Major comments**

[1]. The method sections, especially section 2.3, are very dense and hard to understand without looking at the results. Could they be merged into the Results section, along with where the results/figures are presented?

[2]. Differences between current study and previous literatures need better explanations. In the Discussion section, it was mentioned that the LSW/lNADW transport ratio differed from 53N array observations because of a different water mass definition in this study. But I am having a hard time understanding the explanations. I thought the LSW and lNADW layers were defined using a fixed isopycnal 27.86 $kg/m^3$. In this sense, shouldn't the LSW (26.7 Sv) and lNADW (3.4 Sv) transports derived from the particles equal the Eulerian transports in the corresponding density layer? Does this large transport ratio suggest a model bias in simulating the overflow waters?

A relevant comment is that the authors claimed the 5.7 Sv NADW from the Greenland-Iceland-Scotland ridge was consistent with overflow observations of 6 Sv. However, the NADW transport in the study is mostly contained in LSW layer, whereas the lNADW transport, which should be used to compare with the overflow observations, is as small as 0.6 Sv.

[3]. Finally, I am trying to understand the consistency/difference between diapycnal mass flux inferred from the Lagrangian particles and the classical diapycnal water mass transformation from a Eulerian perspective. I guess the two cannot be compared directly but they should be ultimately linked. For example, observations show that 7 Sv of lighter waters are transformed into denser layers by surface buoyancy loss in the Irminger and Iceland basins, as reported by Petit et al. (2020). Some of these waters might travel across the gyre and reach the boundary current at 53N, which will be counted as part of the diapycnal mass flux discussed in this study. This was only briefly mentioned in the Discussion. I suggest the authors to elaborate a little bit more.

**Minor comments**

[1]. Lines 3-4: Here you mentioned NEADW and DSOW. However, in the manuscript, the sources of NEADW and DSOW transports were not explicitly distinguished and discussed.

[2]. Line 8: It is better to first report the total transport at 53N in the model, i.e. 30 Sv, before quantifying different sources. Also, please specify in the Abstract that "diapycnal mass flux" refers to the diapycnal flux in the boundary.

[3]. Line 19: "a net downwelling *in density space* of upper AMOC water"

[4]. Lines 66-68: I am not sure if I understand this long sentence. What do you mean by "adding transformed water to a major volume of water…"?

[5]. Equation (1): What is "ceil"?

[6]. Lines 122-126: So the water mass definitions are based on mean density, but the particle release density varies on daily time scales, correct?

[7]. Line 130: What do you mean by "the same advection time"?

[8]. Line 172: Are signs or flow directions considered for the cumulative transport? If particles flow into the bin from different directions, then the cumulative transport should be zero.

[9]. Line 179: Is this binned transport (based on point of origin) also converted to the relative transport with respect to the 53N section? Again, are flow directions considered in the binning?

[10]. Lines 191-193: I do not understand how the "volumetric water mass transformations" are calculated here. Please elaborate.

[11]. Lines 254: The 5.7 Sv of NADW from the ridge is mostly in LSW layer. I am not sure why the authors compare this number with the overflow transport observations. Instead, it is the lNADW transport (as small as 0.6 Sv) that should be compared with overflow observations (6 Sv). Please also see my major comment [2].

---

## Author Comment (AC1)

We thank Referee #1 for their encouraging statement and the constructive comments that improved our manuscript. Please find our full response to the comments, and the changes we have made to address these comments, in the attached pdf file. Referee #1 and #2 similarly asked for the clarification of the terms diapycnal mass flux as well as a more detailed explanation for the differences in water mass definitions in observations and the used ocean model. We thank you for these remarks as the explanations were enhanced through this revision.

In the following the review is copied in black with our comments and answers in blue just below each remark.

**Review on "Major sources of North Atlantic Deep Water in the subpolar North Atlantic from Lagrangian analyses in a high–resolution ocean model" by Fröhle et al.**

General Comments

The paper from Fröhle and colleagues investigates the relative contributions of the different sources of the North AtlanticDeep Water (NADW) that exits the Lab- rador Sea at 53$o$N. The manuscript outlines an analysis of Lagrangian particles in a high-resolution model to determine the NADW sources and its pathways and associated timescales from each source to the 53$o$N. The authors detail the interesting finding that within the subpolar North Atlantic the water mass trans- formation towards the density range of the NADW mainly happens through the process of diapycnal mixing (non-convective and convective).

These are interesting results that further our understanding of the high latitude ocean circulation, given all the present discussions of the Labrador Sea and its potential role, at different timescales, in the Atlantic Meridional Overturning Circulation (AMOC). Overall, the paper is generally well organized and clear, however some of the writing could be improved. I am pretty sure that this can be easily addressed by the authors and that might help to improve this already good paper. Thus, I recommend this paper for publication after major revision.

**Specific Comments**

1.   Diapycnal mixing vs deep convection: Although the separation of the two processes might be trivial for most of the readers, I would suggest that the authors clarify a bit better the differences in the manuscript. I believe that the authors aim in distinguishing the diapycnal water mass transformations which are associated with deep convection, thus unstably stratified water masses, from water mass transformations, which are caused by internal diapycnal mixing across stratified density layers. Although in section 2.2 the authors clearly separate the two processes for the particles' categories, I would recommend a better explanation of the two processes in the rest of the manuscript.

An explanatory sentence was added in the section about the particle categories

"If particles increase their density during the experiment from $\sigma_0 < \sigma_{DW}$ to $\sigma_0 \geq \sigma_{DW}$ before reaching 53°N outside of the mixed layer, without contact to the atmosphere, this is referred to as diapycnal mass flux and the particles are classified as $DIA_P$. Else, if the respective density increase occurs within the mixed layer, with contact to the atmosphere, the particles are classified as $ML_P$." Additionally in the entire document it was clarified that we mean internal diapycnal mixing below the mixed layer, without contact to the atmosphere, when we talk about diapycnal mixing.

2.   Guidance to the reader: I found myself wondering too many times at which figure/table should I look for many statements in the manuscript and in particular in section 3. I pointed out a few such examples below, but it is not an exhaustive list. Also, the terminology used in this manuscript regarding the source and the point of origin for each particle category is somehow mixed within the text.

Methods section part 2.1.3 adapted and added to: "For each particle, the trajectory is considered only between the particle's origin, described in detail in the following, and 53°N. Resulting from the definition of the point of origin, the trajectories have varying lengths. In turn these are consequently related to varying transit times. However, all resulting trajectories lie entirely within the NADW density range and within the North Atlantic. The terms source, origin and point of origin are used synonymous in this work."

3.   Discussion (section 4): I feel that this section needs a better structure. It might be helpful to add some subsections to organize the discussion of your results. Many paragraphs are rather large with mixed information and difficult to follow. Furthermore, in many statements a reference to the relevant figure/table is missing. I would also suggest that the discussion begins with a short summary of the goal/methodology of this manuscript.

A short summary has been added to the beginning of the Discussion, which is now divided into several sub-sections. Furthermore, more references to figures and tables are now added.

**Technical Comments**

1.  **Title:** This study is based on the results of one experiment, right? Therefore, I suggest that you change ".. Lagrangian analyses.." to ".. Lag- rangian analysis.." in the title.

Since we use a multitude of techniques to analyze the Lagrangian experiment it is called analyses. We want to clarify here: The experiment or ocean model used here is Viking20x-JRA-OMIP and for the analysis of this model the offline Lagrangian particle experiment is used. This experiment is then analyzed in various ways.

2.  **Line 1:** The North Atlantic Deep Water (NADW) ..

Done

3.  **Line 3:** ... components of NADW (namely..)

Deleted the different classes of NADW since the result of this paper shows that in the model there are different sources of NADW related to similar density classes but not to certain regions as suggested in observations.

4.  **Line 4 and elsewhere:** experiments –> experiment

Done in the entire manuscript.

5.  **Line 5:** "according to the strength of the velocity field", this could be removed

It is valuable information since the amount of particles released is dependent on the velocity field - we keep this information here.

6.  **Line 5:** change "computed" to "traced"

Done

7.  **Line 6-7:** "Water masses were defined ... hydrography field", I don't see the importance of this sentence in the abstract. Consider removing this.

In section 2.1.3 we explain how the NADW is defined. And we deleted this sentence from the abstract.

8.  **Line 13:** Please consider rephrasing "... is hence dominated by the processes of diapycnal mixing and deep convection in the Labrador Sea."

Since the deep convection "only" attributes overall half of the diapycnal mass flux we leave it as is.

9.    **Line 15:** I believe that a short description of the AMOC would be beneficial.

We added an introductory sentence to the introduction: "The Meridional Overturning Circulation (MOC) is the global redistribution system of heat, mass, fresh water and tracers.Water mass transformation from the upper to the lower MOC component associated with deep convective mixing [LabSea 1998, Marshall et al 1999] and diapycnal mixing [Straneo et al. 2006, Katsman et al. 2018, Johnson et al. 2019] is occurring in only few key regions globally. One of them being the highly complex region of the subpolar North Atlantic (SPNA).".

We focus on the origin of deep water and not on the interconnection between overall AMOC and deep water formation through different processes. We discuss the difference between DWBC export and AMOC with our experiment in the Discussion section 4.5 .

10.    **Line 16-18** Please rephrase this sentence in line with the general comment #1. Also, references of Straneo and Katsman are odd here. You could refer to Johnson et al. (2019)1.

Straneo and Katsman talk about the role of diapycnal mixing for densification in the SPNA - we keep the references and add Johnson et al 2019.

11.    **Line 22:** .. of the North Atlantic Deep Water ..

Done

12.    **Line 26-28 & 31-32 & 33:** References are missing.

Done; references are added for the water mass definitions.

13.    **Line 45:** further –> farther

Done

14.    **Line 46:** east Greenland –> East Greenland

Done

15.    **Line 53:** ".. is not finally understood.. " –> still remains unclear

Done

16.    **Line 54:** Consider adding more recent references than Lozier (2012) and Rhein et al. (2013).

Added Yeager et al. 2022.

17.    **Line 60-62:** Please revise this sentence and add references of relevant studies.

Done; references were shifted to this sentence.

18.       **Line 68-69:** "Additionally, the observed .." –> Additionally, the deep convection in the Irminger Sea is more frequently evident in recent obser­vational studies ... or similar.

Matter of taste - we left the sentence as is, but added Rühs et al. 2021 as reference. We additionally added the phrase: " Biastoch et al. 2021 show that the model is reproducing the major, and regional, dynamic properties in the SPNA region, such as the strength and width of the boundary currents, the position, depth and expansion of the mixed layer [Rühs et al. 2021], as well as an AMOC strength comparable to observations. " in section 2.1 .

19.       **Line 75:** ".. and pathways of the single.." –> ".. and pathways of each of the deep water sources..

Changed to : "Subsequently, we present the sources and pathways of each deep water particle category in section ..."

20.       **Line 77:** section 5 is omitted here.

Added

21.       **Line 96:** .. along a section –> at a section

Done

22.       **Line 99-100:** Are the particles released over the whole water column in the vertical?

Yes they are; clarified in section 2.1.1.

23.       **Line 115:** How are the particles traced backward in time? Do you use the velocity/hydrography field averaged over a certain period and repeat this as input for Parcels or do you use the daily snapshots from the release day of each particle and backwards?

Daily snapshots are used for seeding and tracing of the particles. Changed "The daily three-dimensional Eulerian flow and hydrographic fields are used here for the offline Lagrangian particle tracking experiment." to "Here, daily snapshots of the three-dimensional Eulerian flow and hydrographic fields are used for the offline Lagrangian particle tracking experiment." for clarification.

A 4[th] order Runge-Kutta scheme is used to integrate the particles in time as stated in section 2.1.2

24.       **Line 142-144:** Is the inspection of the location of the particle's density change (within or outside the mixed layer) based on a time-mean value of the mixed layer depth?

No, it is based on the MLD that we trace throughout each particle trajectory. Added a sentence for clarification: "To separate $DIA_P$ from $ML_P$ the particle depth is compared to the instantaneous mixed layer depth at the particle

position, which is stored during the experiment along each particle's trajectory (section 2.1.2)."

25.        **Line 155:** Why did you select the 3000 m isobath for the separation between the boundary current and the interior? Please also consider mentioning the reasoning behind this separation.

Changed to "Since, in the SPNA the boundary current sticks to the strong shelf break, the particle is classified as being in the boundary if the underlying bathymetry is shallower than 3000 m in the Labrador Sea, or 2000 m in the remaining SPNA."

The two-part definition has now been introduced, since the 3000 m isobath fits the boundary current in the Labrador Sea rather well, but the 2000 m isobath is a much better approximation for the remaining SPNA. See Figure 1 for a comparison.

26.        **Line 177 and elsewhere:** In my understanding, the point of origin is relevant only for the $DIA_p$ and $ML_p$ particles. If so, please mention this here and keep a consistency between the terms "source" and "point of origin" in the remaining of the manuscript.

"Point of origin" and "source" are used synonymously. It is only relevant for $DIA_P$ and $ML_P$ insofar as the location of it varies. For $DS_P$ and $ISR_P$ the point of origin is fixed by definition. The corresponding paragraph in section 2.1.3, where the point of origin is introduced has been revised. Furthermore, we streamlined these expressions in the whole manuscript.

27.        **Line 178-79:** ".. based on their respective starting points." What do you mean here?

Meant is the point of origin, i.e. the point where the particle enters the NADW within the SPNA. The paragraph has been revised.

28.        **Line 180:** "To determine .. respective basin." –> Rephrase to something like: "To determine the transport contribution of the different basins within the subpolar North Atlantic, we define ...". You could omit the limits of each area that you define and just refer to figure 2 for the definition of the different areas.

For reproducibility we prefer to keep the exact information in the manuscript, however, the limits are moved to the figure caption of Figure 1, where the boundaries are now shown.

29.        **Line 185-186:** You define the travel time of each particle based on the point of origin (point where the particle changes its density). However, in Figure 5 you include the transit time of $DS_p$ and $ISR_p$. Please revise this sentence.

Each particle has a defined point of origin (see also point 26). The time it takes the particle to reach 53°N starting from this point of origin is referred to as "transit time" (this phrase has been streamlined throughout the manuscript to avoid

confusion [i.e. travel time, advection time, … have been replaced]). Thus, transit times can be calculated for all four categories.

30. **Figure 2:** Why did you choose to average the mixed layer depth over the 2000-2019 period and not over 1958-2019 or 2010-2019, which are periods that are discussed earlier in the manuscript? Also, this figure is mainly discussed after Figure 3. Consider changing the order of these figures. If you do so, you could remove the blue dashed lines that you use to define the areas in Figure 2 and add these lines in Figure 1.

The aim here is to give a mean picture of the MLD structure in the SPNA during the period that is relevant for the presented analyses. Particles are released during the period 2010-2019. Most of the $ML_P$ are advected less than 10 years, therefore essentially no particles of this category are circulating within the SPNA prior to 2000.

Since the MLD is subject to strong multi-decadal variability, we did not choose to use an average over the whole model run (1958-2019) to exclude e.g. the period of extensive deep convection during the 1990s, which is not relevant in the context given above.

On the other hand, averaging over the seeding period (2010-2019) would not take into account the period prior to 2010 that is still important for some particles, especially those released in the early 2010s.

Therefore, using the period 2000-2019 gives a mean picture of the MLD structure during the period during which the vast majority of $ML_P$ is circulating in the SPNA.

Also the figure order has been changed as suggested and the blue dashed lines are now included in Figure 1.

31. **Table 1:** Please check the values. The transports/contributions associated with each particle category do not sum up to the total transport (i.e., for NADW: 101% instead of 100%, for LSW: 12.8+7.0+3.4+1.7+1.7 =26.6 Sv instead of 27.7 Sv). There is also an extra parenthesis at 3 column, 4 row.

Table caption of table 1 was adapted to: "The transports are rounded to 0.1 Sv."

32. **Line 200:** I am missing a short introductory paragraph/sentences here. It is a rather rough beginning for the reader. It is not clear in which figure the reader should look.

Done.

33. **Line 283:** "The upper transport.." –> The lighter transport peak ..?

Changed to: "The upper, lighter transport peak is associated with transport peaks around …"

34.    **Line 249:** "... is dominated by diapycnal mass flux and the particle residuum." –> "... is dominated by the $DIA_p$ and $RES_p$ particles.

Done

35.    **Line 207:** ($ML_p$, 7.0 Sv or 26%, Table 1)

Done

36.    **Line 214:** I don't feel this has been fully demonstrated. Perhaps adding the region of high EKE in one of the figures would help.

Added reference to Rieck et al 2019a to clarify and support our finding.

37.    **Line 219:** "... within the boundary current in the Labrador Sea (5.5 Sv, **Table 2**) and Irminger Sea (4.6 Sv, **Table 2**) **at** depths between ..."
Done

38.    **Line 222-224:** I guess that this statement refers to Figure 4 a. However, I don't see the 1000 m isobath.

References to corresponding figures are now added.

39.    **Line 225-229:** References to the associated figure/table are missing. Also, consider adding the values of INADW in Table 2.

Sentence added in the table caption: "Values are given for $\sigma_0 \geq 27.62\ kg\ m^{-3}$ (NADW) and $27.62 \leq \sigma_0 < 27.86\ kg\ m^{-3}$ (LSW), the difference between the two corresponds to the transport associated with $\sigma_0 \geq 27.86\ kg\ m^{-3}$ (INADW).", Table 2 is referenced at the end of the sentence.

40.    **Line 231:**"... single regions in the interior..". What do you mean?

Changed the sentence to : "Contrary to the boundary current the interior does not show as elevated values and a spread over a larger area (Figure 4a)".

41.    **Line 240:** "Boundary Current" –> boundary current

Done

42.    **Line 248:** add a reference to Figure 5b.

Done

43.    **Line 253:** "Transport" –> transport

Done

44.    **Figure 5:** Consider making a new figure for the panels (e-f). You only mention these panels shortly in section 4 (Discussion).

Panels (e-f) are moved to a separate figure (A8) in the appendix, now also including the depth evolution of $DIA_P$ and $ML_P$.

45.    **Line 259 and Figure 5:** The definition of the transit/travel time here differs from $DIA_p$ and $ML_p$, right?

The definition of transit time is consistent, i.e. the time it takes a particle to travel from its point of origin to 53°N. However, the definition of "point of origin" varies among the particle categories.

46.    **Line 267-268:** Please be more concrete here. What do you mean by "due to the importance of interior pathways"?

Changed to "... as relatively more particles tend to be advected through the basin interior compared to the boundary currents."

47.    **Line 275-284:** Why not including a figure showing the major pathways of $RES_p$?

The figure has now been added to the supplementary figures (A2).

48.    **Line 286-289:** Please revise; refer to earlier figure/table to support your statement and guide the reader of what is following in this section.

Done.

Changed to : "We evaluate the changes the water parcels undergo during their spreading routes from their point of origin to the 53°N target section. The evaluation is done for each particle class, except $RES_P$, ordered by the relative contribution of the respective particle class to the transport at 53°N. All particle categories, apart from $RES_P$, show similar water mass property signatures at 53°N."

49.    **300-301:** Do you mean the region south of 53*o*N? Is this still considered the interior of the SPNA?

Changed to "South of 53°N…"

50.    **Line 312-319:** Here, many elements of discussion have already been discussed in section 3.1.2. Consider revising the text.

Shortened and reordered: "Consistent with previous studies, both observational and model–based (Pickart et al., 1997; Marshall and Schott, 1999; Pickart et al., 2002; Cuny et al., 2005; Brandt et al., 2007; MacGilchrist et al., 2020; Georgiou et al., 2021), those mixed layer ($ML_P$) origins contributing majorly to the 53°N transport, are located within the central Labrador Sea and the Western Boundary Current region in the Labrador Sea (Figure 2 b and Table 2). The contribution from the boundary regions exceeds the direct contribution from the interior (Table 2). In agreement with Koelling et al. (2022) the export of these $ML_P$ at 53∘N is between February and April and the transit times between formation and export are only a few months (Figure 5 b)."

51.    **Line 335:** ".. is followed by a continuous decrease in salinity until 53°N." Is this statement verified by a figure?

A figure (A7) has been added to the appendix as support.

52.     **Figure A2:** There are some extra parenthesis in the caption.

Fixed.

53.     **Figure A3 and Figure A4:** Consider adding a title in each panel. Also, mention in the caption of figure A3 that this figure is for the DIA particles.

Done.

54.     **Line 345-346:** Please revise this sentence.

Done; the sentence was rephrased : "With 48 % of the total NADW and LSW transport, the DIA$_P$ represent the majority of NADW (LSW) at 53°N in this experiment (Table 1). This result aligns with the results of Lumpkin et al. (2008) and Marsh et al. (2005), who found that most of the LSW, leaving the SPNA southward, originates from subsurface diapycnal mixing, without contact to the atmosphere, rather than directly from the mixed layer as a result of air–sea fluxes. "

55.     **Line 357-358:** If the water masses are laterally advected within an isopycnal, how a change in density is then possible?

The water masses are laterally advected and due to eddy mixing into the boundary, the particles densify. This process is explained in Brüggemann et al. 2019. The sentence was adapted to : " Based on an idealized model, Brueggemann et al. (2019) showed that densification can also be related to transport of water masses from lower to higher densities. In this case water masses are advected laterally via mesoscale eddies into the boundary current across an isopycnal, leading to a change in density. "

56.     **Line 378-380:** Figure 5(e-f) is only shortly discussed here. I don't see the relevance here. Is the signal of downwelling only related to the particles that originate at the GSR? What about the other particle categories?

Panels (e-f) of Figure 5 are moved to a separate figure in the appendix, now also including the depth evolution of  DIA$_P$ and ML$_P$ . The text was revised.

57.     **Line 394-396:** Please revise.

Sentence was rephrased : "The convection area along with the produced density and volume produced through convection in the Irminger Sea is comparable to the Labrador Sea in the period 2015-2018 [Rühs et al. 2021]."

58.     **Line 405-406:** I guess that here you are referring only to the contribution of ML$_p$. If so, why? What about DIA$_p$?

Changed to overall transport contribution (DIA$_P$+ML$_P$) from Labrador Sea compared to Irminger Sea.

59.     **Line 412-420:** Please add references to the associated figures.

Done

60.     **Line 420-421:** Please revise.

Done and rephrased : "South of 53°N all particle categories feature a cyclonic re-circulation cell around Orphan Knoll, which is in agreement with previous studies [Lavender et al. 2000, Xu et al. 2010]"

61.      **Line 422:** Volume transports of what?

Changed to "Concerning the volume transports of the respective particle classes, our results are only faintly comparable to existing literature."

62.      **Line 438 and elsewhere:** stemming –> originating or similar.

Done

---

## Author Comment (AC2)

We thank Referee #2 for their encouraging statement and the constructive comments that improved our manuscript. Please find our full response to the comments, and the changes we have made to address these comments, in the attached pdf file. Referee #1 and #2 similarly asked for the clarification of the terms diapycnal mass flux as well as a more in detail explanation for the differences in water mass definitions in observations and an ocean model. We thank you for these remarks, as the explanations were enhanced through this revision.

In the following the review is copied in black with our comments and answers in blue just below each remark.

Review on "Major sources of North Atlantic Deep Water in the subpolar North Atlantic from Lagrangian analyses in a high-resolution ocean model" by Fröhle et al.

**Summary**

Fröhle et al. investigated the major sources of the North Atlantic Deep Water transports at the southern exit of the Labrador Sea using Lagrangian particle experiments in a high-resolution ocean model. They quantified contributions from different processes, including diapycnal fluxes (in and out of the mixed layer) and overflow from the Greenland-Iceland-Scotland ridge, to the total deep water transports. For each source, the associated pathways and transit times were discussed.

Overall, I find results from the study quite interesting as they show, in a model, what the subpolar deep western boundary current is composed of from a Lagrangian perspective. The manuscript was overall clearly written and the particle experiments were reasonably designed, supporting the major conclusions. Therefore, I recommend this manuscript for publication after addressing the following comments.

**Major comments**

[1]. The method sections, especially section 2.3, are very dense and hard to understand without looking at the results. Could they be merged into the Results section, along with where the results/figures are presented?

Thank you for this valuable comment. We adapted your proposition in the following way: The subparagraphs are now highlighted with fat letters for the key analysis techniques to enhance the readability and guide and make it easier to find certain techniques in the manuscript. We will keep the separation between methods and results in order to stick to a clear structure of the document.

[2]. Differences between current study and previous literature need better explanations. In the Discussion section, it was mentioned that the LSW/INADW transport ratio differed from 53N array observations because of a different water mass definition in this study. But I am having a hard time understanding the explanations. I thought the LSW and INADW layers were defined using a fixed isopycnal 27.86 $kg/m^1$. In this sense, shouldn't the LSW (26.7 Sv) and INADW (3.4 Sv) transports derived from the particles equal the Eulerian transports in the corresponding density layer? Does this large transport ratio suggest a model bias in simulating the overflow waters? A relevant comment is that the authors claimed the 5.7 Sv NADW from the Greenland-Iceland- Scotland ridge was consistent with overflow observations of 6 Sv. However, the NADW transport in the study is mostly contained in  LSW layer, whereas the INADW transport, which should be used to compare with the overflow observations, is as small as 0.6 Sv.

The LSW / INADW boundary is defined based on the hydrography in the Labrador Sea, assuming that LSW is majorly formed within the Labrador Sea. However, in the presented experiment it is revealed that (at least in the model) LSW is not solely constituted of water masses formed within the Labrador Sea, but there are contributions from different sources and processes. Thus, the used definition is in this context probably insufficient. This has been discussed in the Discussion section.

To better avoid confusion, a paragraph about this problem has been added in the methods section, where the boundary isopycnals are introduced. " The water mass boundaries are defined as the mean density value over the complete model output, covering 1958 through 2019. Contrary to the dynamically defined upper bound of NADW, the definition of the boundary between LSW and INADW is based on the hydrography in the central Labrador Sea [Handmann et al. 2018]. Even though this method works fine with observations and yields the distinguished densities of the three NADW water masses, we show in this study, this does not necessarily hold for a water mass distinction in the classical sense in an ocean model. This is partly related to the unrealistically large diapycnal mixing in regions where dense waters descend topographic slopes, producing lighter water Willebrand et al. (2001). This spurious mixing is dependent on the vertical and horizontal resolution of the ocean model and is a typical model artifact."

Furthermore, within the results section, the terms LSW and INADW are replaced by the corresponding density ranges. The terms themselves are discussed in the Discussion section, where the corresponding section has been moved to the beginning of the Discussion to emphasize the importance of this finding.

As for the overflow transports, the Eulerian estimates reported in this study are given at the sills themselves, not at 53°N. The estimate for the Lagrangian experiment presented is based on the transport assigned to each particle. As explained in the manuscript, the transport is conserved along the particle trajectories. Thus, particles amounting to 5.7 Sv cross the GSR and reach 53°N. The volume is not changed, but due to mixing the density decreases and the particles mostly end up in the LSW layer at 53°N. However, at the GSR itself, the estimate based on the Lagrangian analysis presented and the Eulerian estimates reported from other studies [Biastoch et al. 2021] are comparable.

[3]. Finally, I am trying to understand the consistency/difference between diapycnal mass flux inferred from the Lagrangian particles and the classical diapycnal water mass transformation from a Eulerian perspective. I guess the two cannot be compared directly but they should be ultimately linked. For example, observations show that 7 Sv of lighter waters are transformed into denser layers by surface buoyancy loss in the Irminger and Iceland basins, as reported by Petit et al. (2020). Some of these waters might travel across the gyre and reach the boundary current at 53N, which will be counted as part of the diapycnal mass flux discussed in this study. This was only briefly mentioned in the Discussion. I suggest the authors to elaborate a little bit more.

The water transformed to NADW density at the surface counts as part of the $ML_P$ class. Only particles that are transferred below the mixed layer from the upper AMOC component to the NADW component count as diapycnal transfer in this study. This was specifically clarified in the section 2.1.3 "Categorisation of particles": "If particles increase their density during the experiment, from $\sigma_0 < \sigma_{DW}$ to $\sigma_0 \geq \sigma_{DW}$ , outside of the mixed layer before reaching 53°N, without contact to the atmosphere, this is referred to as diapycnal mass flux and the particles are classified as $DIA_P$. Else, if the respective density increase occurs within the mixed layer, with contact to the atmosphere, the particles are classified as $ML_P$.".  We also state a short version of this in the abstract : "Our experiment shows that, of the 30.1 Sv of NADW passing 53° N on average, the majority is associated with diapycnal mass flux without contact to the atmosphere…. ".

Additionally, Petit et al. (2020) focus on the recent period, where we average over the period 2010-2019.

**Minor comments**

[1]. Lines 3-4: Here you mentioned NEADW and DSOW. However, in the manuscript, the sources of NEADW and DSOW transports were not explicitly distinguished and discussed.

Thank you for this comment. As we take all NADW defined as densities larger than the AMOC density at OSNAP at 53°N to investigate the sources. The results show that we cannot define the origins of LSW, NEADW nor DSOW by solely density in the model. We deleted this separation from the abstract since we do not use it later on. We mention this reasoning also in the Discussion (subsection 4.1) : "To conclude, we find that the density interval with the major transports in NADW at 53°N around $\sigma_0$ = 27.80 kg m$^{-3}$ is not only associated with one source. Instead multiple sources contribute with different relative importance to similar density regimes, though the $DIA_P$ and $ML_P$ dominate ."

[2]. Line 8: It is better to first report the total transport at 53N in the model, i.e. 30 Sv, before quantifying different sources. Also, please specify in the Abstract that "diapycnal mass flux" refers to the diapycnal flux in the boundary.

The entire sentence in the abstract is changed to : "Our experiments show that, of the 30.1 Sv of NADW passing 53° N, the majority is associated with diapycnal mass flux without contact to the atmosphere, accounting for 14.3 Sv (48%), where 6.2 Sv originate from the Labrador Sea, compared to 4.7 Sv from the Irminger Sea."

[3]. Line 19: "a net downwelling *in density space* of upper AMOC water"

Done

[4]. Lines 66-68: I am not sure if I understand this long sentence. What do you mean by "adding transformed water to a major volume of water…"?

Sentence changed to "Newer research has shown that a major volume of water is transformed along the North Atlantic Current path [Desbruyeres et al. 2019]. This water originates from different transformation processes, which are related to different export time scales [Le Bras et al. 2020]. Hence, the very localized deep convection might only be adding a comparably small amount of transformed water to the NADW overall volume."

[5]. Equation (1): What is "ceil"?

Replaced by mathematic symbols for the ceiling function. And added comment under formula to explain.

[6]. Lines 122-126: So the water mass definitions are based on mean density, but the particle release density varies on daily time scales, correct?

Yes; reformulated the corresponding passage ("The particles are subsampled based on their density at their respective release, i.e. only particles with densities $\sigma_0 \geq \sigma_{DW}$ are considered, resulting in a subset of particles.").

Additionally added the sentence "Contrary to the dynamically defined upper bound of NADW, the definition of the boundary between LSW and lNADW is based on the hydrography and, as shown in this study, does not necessarily hold for a water mass distinction in the classical sense." for clarification of the water mass definitions.

[7]. Line 130: What do you mean by "the same advection time"?

For clarification the sentence was changed to: "For each particle, the trajectory is considered only between the particle's origin, described in detail in the following, and 53°N. Resulting from the definition of the point of origin, the trajectories have varying lengths. In turn these are consequently related to varying transit times."

[8]. Line 172: Are signs or flow directions considered for the cumulative transport? If particles flow into the bin from different directions, then the cumulative transport should be zero.

Probability density maps are usually computed following the two possibilities described in Van Sebille et al. 2017. Here, however, the probability density map is weighted with the volume associated with each particle. Particles either pass a box or not. The paragraph has been revised.

[9]. Line 179: Is this binned transport (based on point of origin) also converted to the relative transport with respect to the 53N section? Again, are flow directions considered in the binning?

For the point of origin the binned transport is not converted to relative numbers. Thus, integrating over all grid cells yields the total volume transport at 53°N associated with $DIA_P$ and $ML_P$, respectively. Since only a fixed position is used to create the maps, flow direction is not considered. The paragraph has been revised.

[10]. Lines 191-193: I do not understand how the "volumetric water mass transformations" are calculated here. Please elaborate.

Clarified the explanation paragraph to : "To obtain the **volumetric water mass transformations** the particles are grouped by their water mass properties at 53°N and at their point of origin. The considered properties are $\sigma_0$, absolute salinity $S_A$ and conservative temperature ($\Theta$), with bin sizes of 0.025 kg m$^{-3}$, 0.01 g kg$^{-1}$ and 0.2° C, respectively. These properties were computed from the practical salinity, potential temperature and depth tracked along each trajectory using the TEOS-10 toolbox for Python [TEOS10_2015]. The difference between the volume at 53°N and the volume at the point of origin for $\sigma_0$, $S_A$ and $\Theta$ class then gives the **volumetric water mass transformation**."

[11]. Lines 254: The 5.7 Sv of NADW from the ridge is mostly in LSW layer. I am not sure why the authors compare this number with the overflow transport observations. Instead, it is the lNADW transport (as small as 0.6 Sv) that should be compared with overflow observations (6 Sv). Please also see my major comment [2].

The point of this paper is to show that the classical way to define water masses, based on density intervals, does not hold like in observations in this ocean model (and most probably in most other ocean models). Yes, the major part of NADW passing the GSR is in the lighter region though it is still the overflow. Hence we compare it with the overflow from observations - which density vise is defined differently but dynamically comparable to what we have in the model.